# First-4-week erythrocyte sedimentation rate variability predicts erythrocyte sedimentation rate trajectories and clinical course among patients with pyogenic vertebral osteomyelitis

**Hsiu-Yin Chiang[1], Chih-Wei Chung[1], Chin-Chi Kuo [1,2], Yen-Chun Lo[1], Wei-Shuo Chang[3], Chih-Yu Chi[4,5]** *

**1** Big Data Center, China Medical University Hospital, Taichung, Taiwan, **2** Division of Nephrology, Department of Internal Medicine, China Medical University Hospital, Taichung, Taiwan, **3** Division of Infectious Diseases, Department of Internal Medicine, Asia University Hospital, Taichung, Taiwan, **4** Division of Infectious Diseases, Department of Internal Medicine, China Medical University Hospital, Taichung, Taiwan, **5** School of Medicine, College of Medicine, China Medical University, Taichung, Taiwan

* cychyi@gmail.com

## Abstract

### Background

The trajectory pattern of erythrocyte sedimentation rate (ESR) in patients with pyogenic vertebral osteomyelitis (PVO) and its clinical significance is unclear. We further evaluated whether the first-4-week ESR variability can predict the trajectory pattern, treatment duration and recurrence of PVO.

### Methods

The longitudinal ESR patterns of adults with PVO within the first 6 months were characterized through group-based trajectory modeling (GBTM). The ESR variability within the first 4 weeks was defined using the absolute difference (AD), coefficient of variation, percent change, and slope change. The first-4-week ESR variabilities were analyzed using ordinal logistic regression to predict the 6-month ESR trajectory and using logistic regression to predict treatment duration and recurrence likelihood. The discrimination and calibration of the prediction models were evaluated.

### Results

Three ESR trajectory patterns were identified though GBTM among patients with PVO: Group 1, initial moderate high ESR with fast response; Group 2, initial high ESR with fast response; Group 3, initial high ESR with slow response. Group 3 patients (initial high ESR with slow response) were older, received longer antibiotic treatment, and had more comorbidities and higher recurrence rates than patients in the other two groups. The initial ESR value and ESR − AD could predict the 6-month ESR trajectory. By incorporating the first-4-week ESR variabilities and the clinical features of patients, our models exhibited moderate discrimination performance to predict prolonged treatment (≥12 weeks; *C* statistic, 0.75;

**Data Availability Statement:** All relevant data are within the paper and its Supporting Information files.

**Funding:** This study was supported by China Medical University Hospital (grant number: DMR-108-188; grant number: CRS-106-018). The funder had no role in study design, data collection and analysis, decision to publish, or preparation of the manuscript.

**Competing interests:** The authors have declared that no competing interests exist.

95% confidence interval [CI], 0.70 to 0.81) and recurrence (*C* statistic, 0.69; 95% CI, 0.61 to 0.78).

## Conclusions

The initial ESR value and first-4-week ESR variability are useful markers to predict the treatment duration and recurrence of PVO. Future studies should validate our findings in other populations.

## Introduction

Despite its low incidence in the general population, pyogenic vertebral osteomyelitis (PVO) remains a challenging disease [1]. The management of PVO necessitates a prolonged course of antimicrobial therapy with or without surgical intervention; however, the optimal duration of therapy is unclear and may range from 4 weeks to 3 months [1]. The latest guideline released by the Infectious Diseases Society of America (IDSA) suggests a total duration of 6 weeks of parenteral or highly bioavailable oral antimicrobial therapy for most patients with bacterial vertebral osteomyelitis [2].

Generally, the treatment duration of PVO depends on the patient's response to therapy, as expressed by clinical symptoms and laboratory data [1–3]. In addition to the clinical symptoms, erythrocyte sedimentation rate (ESR) and C-reactive protein (CRP) are commonly used to monitor responses to PVO therapy [1, 2]. In a retrospective study, Lin et al proposed that the lowest ESR ($\geq$20 mm/h) and CRP level ($\geq$5 mg/L) during the treatment course could predict osteomyelitis recurrence and suggested using ESR to determine treatment duration [4]. However, using a threshold-based approach to predict the clinical course or determine the treatment duration for patients with PVO ignores ESR dynamics throughout the disease course [5–8]. In an early study conducted in 1997, Carragee and associates found that patients with an ESR decline (vs pretreatment) of more than 25% after 4-week antibiotic therapy faced a reduced risk of treatment failure [9]. Similarly, as mentioned in the IDSA guideline, patients with PVO exhibiting a 50% reduction from the baseline ESR after 4 weeks of antibiotic therapy rarely experience treatment failure [2]. Therefore, using an ESR time–value curve (trajectory) may be more appropriate to describe and/or predict the clinical course, duration of treatment, and recurrence for patients with PVO.

By leveraging real-world electronic medical record (EMR) data and tracking national health register–based data, we established a comprehensive set of patient-tracked data for PVO at our institution. We aimed to characterize the ESR trajectories among patients with PVO during the first 6 months of disease. We hypothesized that the initial ESR value and variability in ESR during the first 4 weeks can predict the ESR trajectory pattern of patients with PVO and thus indicate the required treatment duration and likelihood of recurrence of PVO.

## Materials and methods

### Study design and study population

This retrospective study employed a study period from December 2002 to May 2014. All adult patients ($\geq$18 years old) who were admitted to China Medical University Hospital (CMUH) during that time and discharged with a diagnosis of discitis, infective spondylitis, infective spondylodiscitis, or PVO were identified from EMR review, and the diagnoses were further

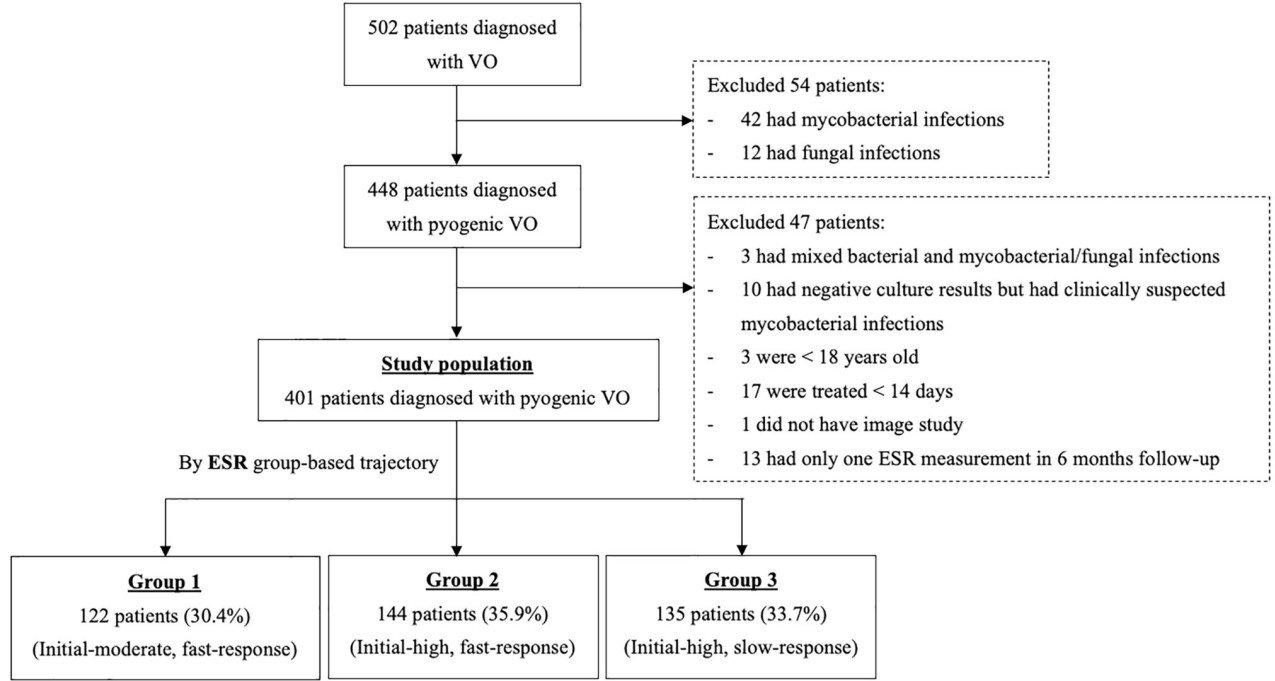

**Fig 1. Study population selection process (*N* = 401). Abbreviations:** ESR, erythrocyte sedimentation rate; VO, vertebral osteomyelitis.

confirmed by infectious disease specialists (CYC and WSC). The inclusion and exclusion criteria and the variable definitions were based on our prior study with some modifications [10]. Briefly, patients were excluded if their antibiotics treatment was shorter than 14 days, if they had a non-hematogenous source of vertebral infection (i.e., artificial implants, prior laminectomy within 1 year, spine penetrating trauma, or decubitus ulcer at the same level of vertebral osteomyelitis), if their vertebral osteomyelitis were caused by mycobacterium, fungus, or Brucella species, or if they did not have any image evidence. In the current study, we further excluded patients who had fewer than two ESR measurements within 6 months following the diagnosis of PVO (Fig 1). In the present study, we linked this PVO cohort with the CMUH clinical research data repository and national death records to obtain biochemistry and mortality data, respectively. This study was approved by the CMUH institutional review board (IRB No: CMUH104-REC2-173 & CMUH105-REC3-068). The corresponding author, CYC, had access to information that could identify individual participants during or after data collection.

## Covariables and outcomes

The covariables included the comorbidities of the patients within 1 year prior to or at the time of PVO diagnosis (e.g., diabetes mellitus, intravenous drug users, liver cirrhosis, end stage renal disease, malignancy, other comorbidities, and Charlson's comorbidity index [CCI], etc.) [11], culture results (i.e., gram-negative pathogens, gram-positive pathogens, polymicrobial microorganisms [≥ 2 different pathogens], or no growth), pre-existing or synchronous infections from distant body sites such as lung (pneumonia), urinary tract, intra-abdomen, bloodstream, or skin and soft tissues within 1 month prior to or at the time of PVO diagnosis, timing of surgical intervention, and biochemical profiles. Baseline biochemical profiles of white blood cell (WBC), c-reactive protein (CRP), ESR, blood urea nitrogen (BUN), serum

creatinine (SCr), and estimated glomerular filtration rate (eGFR, calculated using chronic kidney disease epidemiology [CKD-EPI] collaboration equation) [12] were obtained within 7 days (before or after) of PVO diagnosis. The EMRs were reviewed by two Infectious Diseases Specialists (CYC and WSC) to determine patients' delayed operation and recurrence status. We defined surgical procedures performed within 2 weeks after PVO diagnosis as immediate operations. Patients receiving delayed surgery were defined if they met indications for surgical intervention (e.g., neural compression, spinal instability, or presence of epidural or paravertebral abscesses) at the time of PVO diagnosis but were operated (e.g., discectomy, laminectomy) later than 2 weeks after the diagnosis of PVO. Recurrence was defined as any recurrent symptoms and signs (e.g., fever or pain on the affected site accompanied with abnormal image study or increased inflammatory markers, in the absence of other causes) within 6 months after the completion of the initial antibiotic treatment and received another course of antibiotic treatment based on clinician's decision. In-hospital mortality, 3-month mortality, or 6-months mortality following the PVO diagnosis were obtained by linking to the National Death Registry of Taiwan.

## Variability quantification of ESR

The measurement of ESR was performed at the CMUH central laboratory using a Mixrate-X20 ESR automated analyzer (Vital Diagnostics S.r.l., Via Balzella, Forlì, Italy). All ESR values measured within the first 4 weeks following PVO diagnosis were used to estimate the ESR variability (4-week ESR variability). The initial ESR value was the first recorded within the 4 weeks following PVO diagnosis. The variability of ESR was expressed in the following forms: (1) absolute difference (AD) = final ESR value–first ESR value; (2) coefficient of variation (CV, %) = (standard deviation of ESR/mean of ESR) × 100; (3) percent change (PC, %) = (ESR-AD/first ESR value) × 100; and (4) intercept and slope, which were calculated using a multi-level model including both a random intercept and slope with all ESR measurements clustered within the patients.

## Group-based trajectory modeling of ESR

We used semi-parametric group-based trajectory modeling (GBTM) to characterize the trajectories of ESR values throughout the first 6 months following PVO diagnosis. We used the PROC TRAJs macro function in SAS to fit a semi-parametric mixture model to longitudinal ESR data through the maximum-likelihood method [13–15]. GBTM is designed to identify clusters of individuals (ie, trajectory groups) who have followed a similar developmental trajectory for a variable of interest. Such an approach is useful when the number of subgroups and shape of trajectories in the subgroups are unknown. The number of ESR trajectory groups was determined on the basis of their trajectory shape to facilitate meaningful interpretation. Determination of the ESR trajectories was performed before subsequent analysis.

## Statistical analysis

All statistical analyses were performed using SAS version 9.4 (SAS Institute Inc., Cary, NC, USA) and R version 3.5.1 (R Foundation for Statistical Computing, Vienna, Austria) software. Continuous variables are presented as median and interquartile range (IQR) or mean and standard deviation. Categorical variables are reported as frequency and proportion (%). The associations between ESR trajectory patterns and covariables were analyzed using the Kruskal–Wallis test (nonparametric) or analysis of variance (parametric) for continuous variables and the chi-square test for categorical variables. The test for linear trend from ESR Group 1, 2, to 3 was evaluated by modeling the categorical ESR trajectories as a continuous variable in

simple linear regression models. According to the median values of ESR-AD, we dichotomized the study population (patients with ESR-AD $< -9$ and those $\geq -9$ mm/h). To evaluate the association between the first 4-week ESR variability and 6-month ESR trajectory, we included the initial ESR value and ESR-AD ($< -9$ vs. $\geq -9$) as the main predictors and the three ESR trajectory patterns as response variables in an ordinal logistic regression model. The proportional odds assumption of the ordinal logistic regression model was evaluated using a graphical method. Furthermore, variables that were statistically significant in the univariable analysis (polymicrobial culture, DM, ESRD, malignancy, CCI; Table 1) or were clinically relevant (CRP, WBC, eGFR) were included in the multivariable analysis. By incorporating the first-4-week ESR variabilities, initial CRP, and other potential confounders, we built several models using logistic regression to predict prolonged PVO treatment ($\geq 12$ weeks) and recurrence. In addition, we proposed a directed acyclic graph (DAG) to illustrate the potential causal relationship between the first-4-week ESR variabilities and poor prognosis (S1 Fig). The discriminative performance of these serial models was assessed using a receiver operating characteristic curve and *C* statistic. The calibration performance was verified through a calibration plot and the Hosmer–Lemeshow goodness-of-fit test. All analyses were two tailed, and $P < 0.05$ was considered significant.

## Results

Of the 502 patients with a discharge diagnosis of vertebral osteomyelitis, 401 patients met the current inclusion criteria and were included in the final analysis. According to all available ESR measurements of the included patients in the first 6 months following PVO diagnosis, the ESR trajectories were categorized into three distinct groups through GBTM (Figs 1 and 2). Group 1 (initially moderate, fast response) comprised 122 patients (30.4%) whose initial ESR values were moderately elevated ($<60$ mm/h) before recovering rapidly, Group 2 (initially high, fast response) comprised 144 patients (35.9%) with high initial ESR values ($\geq 60$ mm/h) that responded rapidly to treatment, and Group 3 (initially high, slow response) comprised 135 patients (33.7%) with initially high but slowly recovering ESR values (Fig 2). Contrarily, the trajectories of other inflammatory markers, such as CRP level and white blood cell (WBC) count, did not show distinct patterns with a balanced sample size (S2 Fig).

The patients in Group 3 (median age, 65.6 years) were significantly older than those in Group 1 (57.8 years) and Group 2 (63.2 years) (*P* for trend = 0.002) (Table 1). Compared with Group 1 (27.9%) and Group 2 (32.6%) patients, the patients in Group 3 were more likely to be female (42.2%) (*P* for trend = 0.015). Similarly, with regard to co-morbidities and microbiological data, significantly higher proportions of Group 3 patients had diabetes (49.6%, *P* for trend $< 0.001$), end-stage renal disease (11.9%, *P* for trend = 0.007), malignancy (15.6%, *P* for trend = 0.001), and polymicrobial infection (13.3%, *P* for trend = 0.013) than those of the other two groups. The mean Charlson's comorbidity index was $1.09 \pm 1.69$ for Group 1, $1.47 \pm 1.91$ for Group 2, and $2.04 \pm 2.43$ for Group 3, which was significantly different (*P* for trend $< 0.001$). Considering the biochemical data, the values of the inflammatory markers (WBCs, CRP, and ESR) at the time of PVO diagnosis were significantly higher for the Group 2 and Group 3 patients (*P* for trend $< 0.001$) than those of Group 1. Similarly, patients in Group 2 and Group 3 were more likely to have impaired renal function than those in Group 1 (*P* for trend $< 0.001$). Group 3 patients were more likely to have thoracic (29.6%, *P* for trend = 0.024) and multi-site PVO (*P* for trend = 0.009) than those in Group 1 and Group 2 (Table 1).

The reported frequency of abscess formation was lower for Group 1 patients (73%) than for Group 2 and Group 3 patients. Over half of the patients in each group received immediate

**Table 1. Baseline demographics and clinical characteristics among patients with pyogenic vertebral osteomyelitis (N = 401).**

| Variables | ALL (N = 401) | 1. Initial-Moderate Fast-Response (N = 122) | 2. Initial-High Fast-Response (N = 144) | 3. Initial-High Slow-Response (N = 135) | P-value[a] | P-value for trend[c] |
|---|---|---|---|---|---|---|
| **Age at index date (year)**, median (IQR) | 63.1 (51.6, 71.4) | 57.8 (47.3, 70.7) | 63.2 (51.5, 71.3) | 65.6 (54.9, 73.4) | 0.007 | 0.002 |
| **Female**, n (%) | 138 (34.4) | 34 (27.9) | 47 (32.6) | 57 (42.2) | 0.046 | 0.015 |
| **Culture**, n (%) | | | | | | |
| Gram-negative pathogen | 91 (22.7) | 28 (23.0) | 34 (23.6) | 29 (21.5) | 0.022 | 0.013 |
| Gram-positive pathogen | 217 (54.1) | 64 (52.5) | 81 (56.2) | 72 (53.3) | | |
| Polymicrobial | 29 (7.2) | 4 (3.3) | 7 (4.9) | 18 (13.3) | | |
| No growth | 64 (16.0) | 26 (21.3) | 22 (15.3) | 16 (11.9) | | |
| **Comorbidity**, n (%) | | | | | | |
| Diabetes mellitus | 145 (36.2) | 24 (19.7) | 54 (37.5) | 67 (49.6) | < 0.001 | < 0.001 |
| Intravenous drug users | 34 (8.5) | 15 (12.3) | 13 (9.0) | 6 (4.4) | 0.075 | 0.024 |
| Liver cirrhosis | 48 (12.0) | 15 (12.3) | 15 (10.4) | 18 (13.3) | 0.748 | 0.781 |
| End stage renal disease | 33 (8.2) | 3 (2.5) | 14 (9.7) | 16 (11.9) | 0.017 | 0.007 |
| Malignancy | 35 (8.7) | 5 (4.1) | 9 (6.2) | 21 (15.6) | 0.002 | 0.001 |
| Myocardial infarction | 2 (0.5) | 1 (0.8) | - | 1 (0.7) | 0.567 | 0.954 |
| Congestive heart failure | 30 (7.5) | 7 (5.7) | 8 (5.6) | 15 (11.1) | 0.144 | 0.095 |
| Peripheral vascular disease | 7 (1.7) | 1 (0.8) | 5 (3.5) | 1 (0.7) | 0.142 | 0.914 |
| Cerebrovascular disease | 24 (6.0) | 5 (4.1) | 8 (5.6) | 11 (8.2) | 0.379 | 0.169 |
| Dementia | 11 (2.7) | 2 (1.6) | 5 (3.5) | 4 (3.0) | 0.648 | 0.530 |
| Chronic pulmonary disease | 21 (5.2) | 6 (4.9) | 10 (6.9) | 5 (3.7) | 0.470 | 0.638 |
| Connective tissue disease, rheumatologic disease | 3 (0.8) | 1 (0.8) | - | 2 (1.5) | 0.355 | 0.513 |
| Peptic ulcer disease | 61 (15.2) | 19 (15.6) | 17 (11.8) | 25 (18.5) | 0.293 | 0.484 |
| Mild liver disease | 46 (11.5) | 16 (13.1) | 15 (10.4) | 15 (11.1) | 0.779 | 0.626 |
| Hemiplegia or paraplegia | 6 (1.5) | 3 (2.5) | 3 (2.1) | - | 0.207 | 0.100 |
| Metastatic solid tumor | 6 (1.5) | 1 (0.8) | 1 (0.7) | 4 (3.0) | 0.226 | 0.149 |
| AIDS/HIV | 2 (0.5) | 1 (0.8) | 1 (0.7) | - | 0.594 | 0.345 |
| **Charlson's comorbidity score,** | | | | | | |
| Median (IQR) | 1.55 ± 2.07 | 1.09 ± 1.69 | 1.47 ± 1.91 | 2.04 ± 2.43 | 0.001 | < 0.001 |
| 0 | 172 (42.9) | 60 (49.2) | 66 (45.8) | 46 (34.1) | 0.005 | < 0.001 |
| 1 | 81 (20.2) | 31 (25.4) | 26 (18.1) | 24 (17.8) | | |
| 2 | 55 (13.7) | 16 (13.1) | 19 (13.2) | 20 (14.8) | | |
| ≥ 3 | 93 (23.2) | 15 (12.3) | 33 (22.9) | 45 (33.3) | | |
| **Preexisting or synchronous infection**, n (%) | | | | | | |
| Pneumonia | 48 (12.0) | 12 (9.8) | 17 (11.8) | 19 (14.1) | 0.577 | 0.295 |
| Urinary tract | 99 (24.7) | 24 (19.7) | 32 (22.2) | 43 (31.9) | 0.054 | 0.022 |
| Intra-abdomen | 32 (8.0) | 6 (4.9) | 14 (9.7) | 12 (8.9) | 0.316 | 0.253 |
| Bloodstream | 62 (15.5) | 13 (10.7) | 22 (15.3) | 27 (20.0) | 0.117 | 0.038 |
| Skin and soft tissue | 24 (6.0) | 5 (4.1) | 11 (7.6) | 8 (5.9) | 0.479 | 0.559 |
| **Biochemical profiles for diagnosis date ± 7 days**, median (IQR) | | | | | | |
| WBC * 10³, cells/mL | 10.9 (8.0, 14.4) | 9.3 (6.9, 12.9) | 11.8 (8.7, 15.1) | 11.5 (8.9, 14.5) | < 0.001 | 0.002 |
| WBC >10,200 cells/mL, n (%) | 206 (55.7) | 50 (43.5) | 78 (61.4) | 78 (60.9) | 0.007 | 0.007 |
| CRP, mg/dL | 10.9 (4.9, 18.2) | 6.1 (3.1, 12.1) | 11.1 (6, 19.6) | 15.4 (6.6, 25.0) | < 0.001 | < 0.001 |
| ESR, mm/hour | 81 (61, 100) | 62 (44, 75) | 91 (72.5, 104.5) | 90 (78, 105) | < 0.001 | < 0.001 |
| BUN, mg/dL | 17 (11, 27) | 14.5 (11, 19) | 17 (10, 27) | 21 (13, 37.8) | < 0.001 | < 0.001 |
| SCr, mg/dL | 0.9 (0.7, 1.2) | 0.8 (0.7, 1) | 0.9 (0.7, 1.2) | 1.1 (0.8, 1.8) | < 0.001 | < 0.001 |

(*Continued*)

**Table 1.** (Continued)

| Variables | ALL (N = 401) | 1. Initial-Moderate Fast-Response (N = 122) | 2. Initial-High Fast-Response (N = 144) | 3. Initial-High Slow-Response (N = 135) | P-value[a] | P-value for trend[c] |
|---|---|---|---|---|---|---|
| eGFR, mL/min/1.73 m$^2$ | 81.7 (51, 100.8) | 91.3 (71.9, 105.4) | 83.5 (57.1, 101.4) | 64.3 (34.4, 87.7) | < 0.001 | < 0.001 |
| **Location of PVO**, n (%) | | | | | | |
| Cervical spine | 44 (11.0) | 17 (13.9) | 15 (10.4) | 12 (8.9) | 0.419 | 0.199 |
| Thoracic spine | 90 (22.4) | 22 (18.0) | 28 (19.4) | 40 (29.6) | 0.047 | 0.024 |
| Lumbar spine | 290 (72.3) | 84 (68.9) | 112 (77.8) | 94 (69.6) | 0.186 | 0.933 |
| Sacral spine | 68 (17.0) | 22 (18.0) | 24 (16.7) | 22 (16.3) | 0.927 | 0.714 |
| Multiple vertebral | 86 (21.4) | 18 (14.8) | 30 (20.8) | 38 (28.1) | 0.032 | 0.009 |
| **Abscess**, n (%) | | | | | | |
| Epidural abscess | 156 (38.9) | 47 (38.5) | 44 (30.6) | 65 (48.1) | 0.011 | 0.097 |
| Paraspinal abscess | 134 (33.4) | 36 (29.5) | 55 (38.2) | 43 (31.9) | 0.292 | 0.725 |
| Psoas abscess | 88 (21.9) | 17 (13.9) | 43 (29.9) | 28 (20.7) | 0.007 | 0.219 |
| Any abscess | 321 (80.0) | 89 (73.0) | 122 (84.7) | 110 (81.5) | 0.050 | 0.098 |
| Abscess without drainage or operation | 75 (18.7) | 25 (20.5) | 26 (18.1) | 24 (17.8) | 0.830 | 0.583 |
| **Treatment**, n (%) | | | | | | |
| Immediate operation | 206 (51.4) | 64 (52.5) | 74 (51.4) | 68 (50.4) | 0.946 | 0.738 |
| Delayed operation | 65 (16.2) | 9 (7.4) | 27 (18.8) | 29 (21.5) | 0.005 | 0.002 |
| Drainage without operation | 21 (5.2) | 8 (6.6) | 6 (4.2) | 7 (5.2) | 0.683 | 0.638 |
| Medical treatment alone | 109 (27.2) | 41 (33.6) | 37 (25.7) | 31 (23.0) | 0.141 | 0.058 |
| **Total duration of antibiotics therapy** | | | | | | |
| Days, mean ± SD[b] | 105.2 ± 78.2 | 81.7 ± 54.9 | 105.2 ± 66.0 | 126.4 ± 99.7 | < 0.001 | < 0.001 |
| <6 weeks | 60 (15.0) | 24 (19.7) | 21 (14.6) | 15 (11.1) | < 0.001 | < 0.001 |
| 6–11 weeks | 138 (34.4) | 59 (48.3) | 35 (24.3) | 44 (32.6) | | |
| ≥ 12 weeks | 203 (50.6) | 39 (32.0) | 88 (61.1) | 76 (56.3) | | |
| **Outcomes**, n (%) | | | | | | |
| 6-month recurrence | 53 (14.1) | 9 (7.7) | 20 (15.0) | 24 (19.0) | 0.037 | 0.011 |
| In-hospital mortality | 23 (5.7) | 5 (4.1) | 10 (6.9) | 8 (5.9) | 0.606 | 0.545 |
| 3-month mortality | 27 (6.7) | 8 (6.6) | 10 (6.9) | 9 (6.7) | 0.991 | 0.975 |
| 6-month mortality | 37 (9.2) | 10 (8.2) | 13 (9.0) | 14 (10.4) | 0.830 | 0.546 |

**Abbreviations:** BUN, blood urea nitrogen; CRP, C-reactive protein; eGFR, estimated glomerular filtration rate; ESR, erythrocyte sedimentation rate; IQR, interquartile range; NA, not applicable; PVO, pyogenic vertebral osteomyelitis; SCr, serum creatinine; SD, standard deviation; WBC, white blood cell count.

[a]. P-values are calculated by Kruskal-Wallis test for continuous variables and chi-square test for categorical variables.

[b]. The treatment duration was significantly different between the first and the second trajectories (Tukey's test p = 0.034) and between the first and the third trajectories (P < 0.001). The difference was not significant between the second and the third trajectories (P = 0.055).

[c]. P-value for trend was calculated by Spearman's correlation test for continuous variables, Cochran-Armitage Trend Test for binary variables, and simple linear regression for variables with more than 3 categories.

surgical intervention (Table 1); however, the patients in Group 3 tended to undergo the operation in a more delayed manner (21.5%) than patients in the other two groups (7.4% and 18.8% for Group 1 and Group 2, respectively). Furthermore, the mean duration of antibiotic treatment was significantly longer in Group 3 patients than in Group 1 patients (126.4 vs 81.7 days; Tukey's test $P < 0.001$) and Group 2 patients (126.4 vs 105.2 days; $P = 0.055$). Compared with the other two groups, significantly more PVO recurrence was observed among Group 3 patients (19.0%, $P$ for trend = 0.011). The in-hospital, 3-month, and 6-month mortality rate did not differ significantly among these three patient groups.

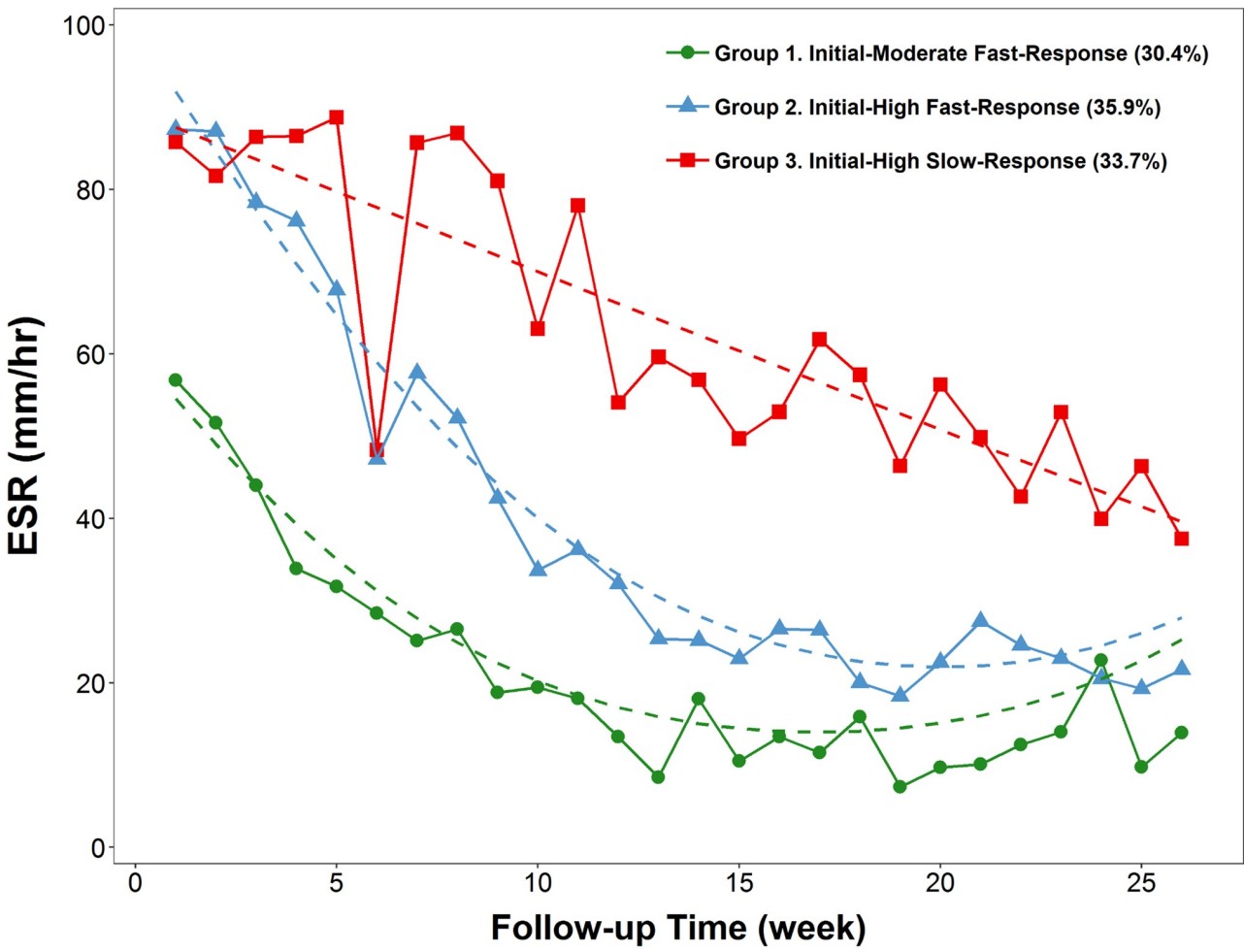

**Fig 2. Group-based trajectory modeling of 6-month erythrocyte sedimentation rate (ESR) among patients with pyogenic vertebral osteomyelitis.**

The median of the initial ESR value (56 mm/h) of the patients in Group 1 was the lowest among the three groups; however, the medians of the initial ESR (88 mm/h for both groups) were identical for Group 2 and Group 3 patients. The medians of ESR-AD and ESR-PC exhibited an increasing trend from Group 1 (−17 and −33.3%, respectively) to Group 2 (−11 and −13%, respectively) to Group 3 (3.5% and 3.6%, respectively; *P* for trend < 0.001). Correspondingly, the other variability measure, such as slope, demonstrated a similar increasing pattern from Group 1 to Group 3 patients. The median time intervals from the first ESR value to the last within the first 4 weeks were similar for the three ESR trajectory groups (19.5 to 22 days) (Table 2).

To evaluate the performance of ESR variability in predicting ESR trajectory, we selected the initial ESR value and ESR-AD as main ESR variability measures because these two were readily applicable and are easy to comprehend (Table 3). For each unit increase in initial ESR, the odds of being in ESR trajectory Group 3 compared with the other two trajectory groups were 6% higher (95% confidence interval [CI], 5% to 7%). Similarly, this interpretation can be extended to compare the patients of trajectory Groups 2 and 3 with the patients in Group 1 (Table 3, Model 1–1). For patients with an ESR-AD decline of less than 9 mm/h, the odds of being in ESR trajectory Group 3 compared with those of being in either other group were 12.5

**Table 2. Variability of erythrocyte sedimentation rate (ESR) during the first 4 weeks following pyogenic vertebral osteomyelitis diagnosis (N = 342), based on ESR trajectory pattern.**

| Variables, median (IQR) | ALL (N = 342) | 1. Initial-Moderate Fast-Decay (N = 115) | 2. Initial-High Fast-Decay (N = 121) | 3. Initial-High Slow-Decay (N = 106) | P-value for trend[a] |
|---|---|---|---|---|---|
| Initial ESR value, mm/h | 78 (55, 95.8) | 56 (38.5, 71) | 88 (73, 103) | 88 (73.2, 103.8) | < 0.001 |
| ESR–AD[b], mm/h | -9 (-27.8, 8) | -17 (-38.5, -1) | -11 (-29, 8) | 3.5 (-13.5, 15.8) | < 0.001 |
| ESR–CV[c], % | 21.2 (10.7, 35) | 30.3 (17.6, 45.5) | 19.4 (9.2, 30.6) | 16 (7.1, 26.9) | < 0.001 |
| ESR–PC[d], % | -12.7 (-36.4, 11.3) | -33.3 (-53.7, -2.5) | -13.0 (-30.2, 10.9) | 3.6 (-12.8, 20.5) | < 0.001 |
| Intercept[e], mm/h | 79.6 (62.3, 93.4) | 61.6 (48.8, 71.1) | 87.9 (73.2, 96.9) | 87.5 (72.2, 99.4) | < 0.001 |
| Slope[e], mm/h | -0.5 (-0.9, 0) | -0.8 (-1.1, -0.5) | -0.4 (-0.9, -0.1) | -0.1 (-0.5, 0.3) | < 0.001 |
| Days from initial to last measure | 21 (15, 25) | 21 (17, 25) | 22 (17, 25) | 19.5 (14, 25) | 0.151 |

**Abbreviations:** AD, absolute difference; CV, coefficient of variation; ESR, erythrocyte sedimentation rate; IQR, interquartile range; PC, percent change; PVO, pyogenic vertebral osteomyelitis.

[a]. P-values for trend are calculated by Spearman's correlation.

[b]. Absolute difference between the first ESR value and the last within the first 4 weeks following PVO diagnosis.

[c]. Coefficient of variation = (Standard deviation/mean) x 100.

[d]. Percent change from the first ESR value to the last within the first 4 weeks following PVO diagnosis.

[e]. Intercept and slope were calculated using a multilevel model including both a random intercept and slope with all ESR measurements clustered within the patients.

times greater (95% CI, 7.4 to 22.0). Correspondingly, the odds for the combined ESR trajectories Group 2 and 3 were 12.5 times greater compared with the odds for trajectory Group 1 (Table 3, Model 1–1). After demographic variables, comorbidities, and baseline eGFR were added in Model 1–2 and polymicrobial infection was added in Model 1–3, the regression

**Table 3. Prediction of erythrocyte sedimentation rate (ESR) trajectory group (Groups 1, 2, 3) through ordinal logistic regression based on ESR within 4 weeks of pyogenic vertebral osteomyelitis diagnosis.**

| Variables | Crude OR (95% CI) | Model 1–1 (AIC: 583.9) [b] | | Model 1–2 (AIC: 544.2) [c] | | Model 1–3 (AIC: 545.5) [d] | |
|---|---|---|---|---|---|---|---|
| | | β | Proportional OR (95% CI) | β | Proportional OR (95% CI) | β | Proportional OR (95% CI) |
| Initial ESR, mm/h | 1.03 (1.03, 1.04) | 0.057 | 1.06 (1.05, 1.07) | 0.056 | 1.06 (1.05, 1.07) | 0.056 | 1.06 (1.05, 1.07) |
| ESR–AD ≥ −9 mm/h [a] | 2.90 (1.94, 4.36) | 2.528 | 12.53 (7.35, 21.96) | 2.578 | 13.18 (7.42, 24.17) | 2.563 | 12.98 (7.31, 23.82) |
| Age, year | 1.02 (1.01, 1.04) | | | 0.005 | 1.01 (0.98, 1.03) | 0.006 | 1.01 (0.98, 1.03) |
| Male | 0.62 (0.42, 0.91) | | | -0.404 | 0.67 (0.40, 1.11) | -0.424 | 0.65 (0.39, 1.09) |
| Diabetes mellitus | 2.63 (1.80, 3.88) | | | 0.441 | 1.55 (0.94, 2.57) | 0.427 | 1.53 (0.93, 2.54) |
| ESRD | 2.38 (1.25, 4.66) | | | -0.248 | 0.78 (0.25, 2.38) | -0.330 | 0.72 (0.23, 2.23) |
| Malignancy | 3.21 (1.63, 6.56) | | | 1.077 | 2.94 (1.11, 8.15) | 1.012 | 2.75 (1.02, 7.73) |
| eGFR, mL/min/1.73 m² | 0.98 (0.98, 0.99) | | | -0.014 | 0.99 (0.98, 1.00) | -0.014 | 0.99 (0.98, 1.00) |
| CCI ≥ 3 | 2.42 (1.57, 3.16) | | | -0.287 | 0.75 (0.38, 1.46) | -0.262 | 0.77 (0.39, 1.50) |
| Polymicrobial infection | 3.43 (1.63, 7.61) | | | | | 0.420 | 1.52 (0.58, 4.09) |

**Abbreviations:** AD, absolute difference; AIC, Akaike Information Criterion; CCI, Charlson's comorbidities index; CI, confidence interval; eGFR, estimated glomerular filtration rate; ESR, erythrocyte sedimentation rate; ESRD, end-stage renal disease; OR, odds ratio; PVO, pyogenic vertebral osteomyelitis.

[a]. We used the median value of ESR–AD as the cutoff value.

[b]. In Model 1–1, initial ESR and ESR–AD were included as predicting variables and three ESR groups as the multilevel response variable.

[c]. In Model 1–2, initial ESR, ESR–AD, demographic information (age, gender), baseline comorbidities (diabetes, ESRD, malignancy, and CCI) and baseline eGFR were included as predicting variables and three ESR groups as the multilevel response variable.

[d]. In Model 1–3 (full model), initial ESR, ESR–AD, demographic information (age, gender), baseline comorbidities (diabetes, ESRD, malignancy, and CCI), baseline eGFR, and polymicrobial infection were included as predicting variables and three ESR groups as the multilevel response variable.

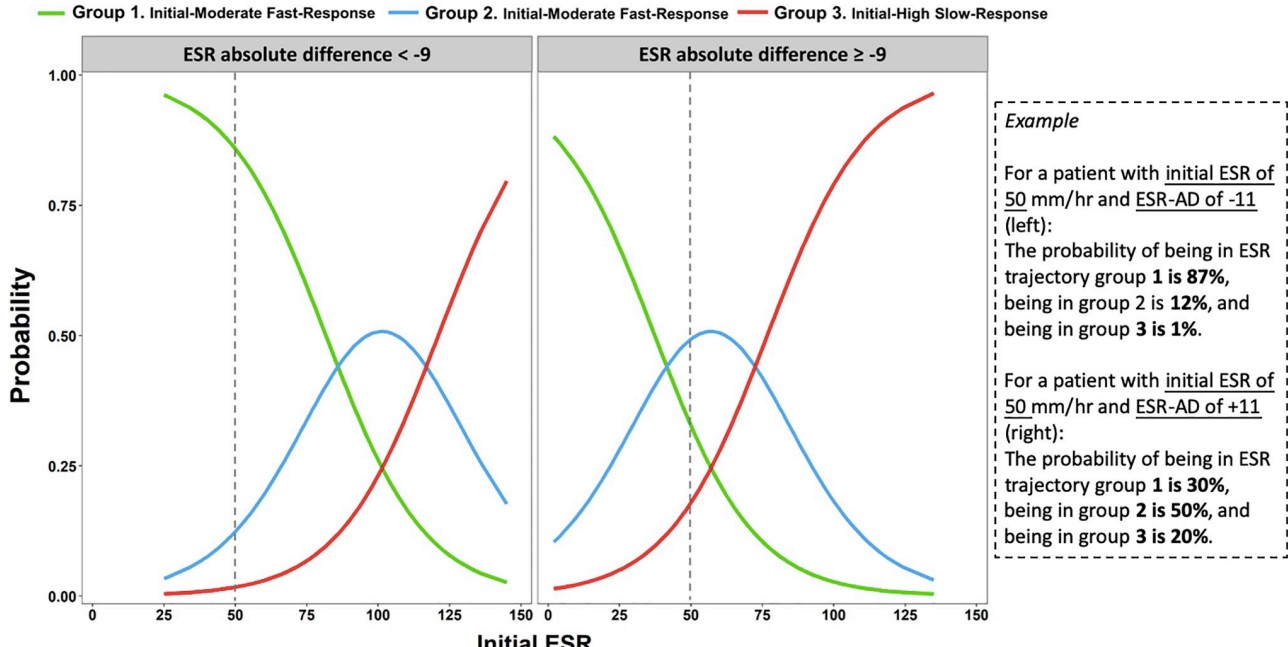

**Fig 3. Predictive probabilities for erythrocyte sedimentation rate (ESR) trajectories based on the initial ESR value and the absolute difference[a] in ESR within 4 weeks of pyogenic vertebral osteomyelitis diagnosis.**

coefficients and odds ratios of the initial ESR and ESR-AD changed subtly (Table 3). Based on the initial ESR value and binary ESR-AD, we again plotted the predicted probability of patients with PVO being classified into one of the three ESR trajectory groups (Fig 3).

Compared with patients with a treatment duration <12 weeks, those treated ≥12 weeks had significantly higher initial ESR value (median, 71.5 vs 85.5 mm/h) and less ESR variation over the first 4 weeks after PVO diagnosis (median ESR-CV, 25.2% vs 17.6%) (S1 Table). Based on the initial ESR and ESR-AD (Model 2–1), the discriminative performance (*C* statistic) for predicting a prolonged treatment (≥12 weeks) was moderate (0.63; 95% CI, 0.57 to 0.69), but it increased to 0.65 (95% CI, 0.59 to 0.71, Model 2–2) after ESR-CV was added to the model. Since the delayed operation and the presence of abscess could influence patients' ESR levels and confound the association between ESR level and outcome (S2 and S3 Tables), we included these variables along with the clinical information, such as the comorbidities, delayed surgery, and abscess formation in the full model. The *C* statistic of the full model (Model 2–3) indicated moderate-to-good discrimination (0.75; 95% CI, 0.70 to 0.81) (Table 4 and Fig 4). Furthermore, no 4-week ESR variability measure was significantly associated with the 6-month recurrence of PVO (S1 Table). The discriminative performance of the initial ESR and 4-week ESR variability, especially ESR-AD and ESR-CV, in predicting 6-month PVO recurrence was poor (Models 3–1 and 3–2); however, their discriminative performance was improved with the addition of clinical information (Model 3–3; *C* = 0.69; 95% CI, 0.61 to 0.78) (Table 5 and S3 Fig). The calibration performance of Models 2–3 and 3–3 are illustrated in S4 and S5 Figs.

## Discussion

Several findings deserved to be highlighted in the present study. First, we evaluated the trajectories of three inflammatory markers (WBC, CRP, and ESR) among patients with PVO;

**Table 4. Prediction of prolonged treatment duration ($\geq$12 weeks) using logistic regression model based on erythrocyte sedimentation rate within 4 weeks of pyogenic vertebral osteomyelitis diagnosis.**

| Variables | Crude OR (95% CI) | Model 2–1 (AIC: 463.3) [b] | | Model 2–2 (AIC: 460.1) [c] | | Model 2–3 (AIC: 373.2) [d] | |
|---|---|---|---|---|---|---|---|
| | | β | Adjusted OR (95% CI) | β | Adjusted OR (95% CI) | β | Adjusted OR (95% CI) |
| Initial ESR, mm/h | 1.01 (1.01, 1.02) | 0.017 | 1.02 (1.01, 1.03) | 0.013 | 1.01 (1.00, 1.02) | 0.014 | 1.01 (1.00, 1.03) |
| ESR–AD $\geq$ −9 mm/h [a] | 1.05 (0.69, 1.60) | 0.381 | 1.46 (0.92, 2.34) | 0.108 | 1.11 (0.66, 1.88) | -0.122 | 0.89 (0.46, 1.68) |
| ESR–CV | 0.98 (0.96, 0.99) | | | -0.017 | 0.98 (0.97, 1) | -0.020 | 0.98 (0.96, 1.00) |
| Initial CRP, mg/dL | 1.02 (0.99, 1.04) | | | | | -0.013 | 0.99 (0.96, 1.02) |
| Age, year | 1.02 (1.00, 1.03) | | | | | 0.019 | 1.02 (1.00, 1.04) |
| Male | 0.95 (0.63, 1.44) | | | | | 0.455 | 1.58 (0.90, 2.81) |
| Diabetes mellitus | 1.17 (0.78, 1.76) | | | | | -0.098 | 0.79 (0.46, 1.35) |
| ESRD | 0.80 (0.39, 1.63) | | | | | -1.508 | 0.22 (0.06, 0.79) |
| Malignancy | 1.73 (0.86, 3.63) | | | | | 1.450 | 4.26 (1.44, 13.91) |
| Any abscess | 1.71 (1.04, 2.83) | | | | | 0.672 | 1.96 (0.97, 4.08) |
| Delayed operation | 1.46 (0.86, 2.52) | | | | | 0.524 | 1.69 (0.79, 3.73) |
| eGFR, mL/min/1.73 m$^2$ | 0.99 (0.99, 1.00) | | | | | -0.020 | 0.98 (0.97, 0.99) |
| CCI $\geq$ 3 | 0.80 (0.50, 1.27) | | | | | -1.061 | 0.35 (0.16, 0.74) |
| *C* statistic (95% CI) | | 0.63 (0.57–0.69) | | 0.65 (0.59–0.71) | | 0.75 (0.70–0.81) | |

Abbreviations: AD, absolute difference; AIC, Akaike Information Criterion; CCI, Charlson's comorbidities index; CI, confidence interval; CRP, C-reactive protein; CV, coefficient of variation; eGFR, estimated glomerular filtration rate; ESR, erythrocyte sedimentation rate; ESRD, end-stage renal disease; OR, odds ratio; PVO, pyogenic vertebral osteomyelitis.

[a]. We used the median value of ESR–AD as the cutoff value.

[b]. In Model 2–1, initial ESR, ESR–AD were included as predicting variables and prolonged treatment duration ($\geq$12 weeks) as the dichotomous response variable.

[c]. In Model 2–2, initial ESR, ESR–AD, and ESR–CV were included as predicting variables and prolonged treatment duration ($\geq$12 weeks) as the dichotomous response variable.

[d]. In Model 2–3 (full model), initial ESR, ESR–AD, ESR–CV, initial CRP, demographic information (age, gender), baseline comorbidities (diabetes, ESRD, malignancy, and CCI), presence of abscess, delayed operation, and baseline eGFR were included as predicting variables and prolonged treatment duration ($\geq$12 weeks) as the dichotomous response variable.

however, neither the CRP nor WBC trajectories appropriately reflected the clinical course of PVO. A considerable proportion of patients (26%−65%) with osteomyelitis have normal or slightly elevated WBC counts at the time of admission [16–18]. Similarly, 44.3% of our patients also had normal initial WBC counts ($\leq$10,200 cells/mL) at the time of PVO diagnosis. Additionally, WBC trajectories did not show significantly different pattern and 73.5% of our study population had flat pattern of WBC (Group 1, S2B Fig). From these clinical observations, it is clear that WBC counts are not suitable as a biological maker in follow-up for patients with PVO. Although the roles of both CRP and ESR in the monitoring of the treatment response for osteomyelitis have been extensively evaluated [4, 9, 16, 18–25], it remains unclear which is more appropriate for monitoring the clinical course of PVO. With appropriate antibiotic therapy, the inflammatory processes induced by osteomyelitis decrease in intensity progressively. The inflammatory marker CRP may return to its normal level within 1 week [16, 18, 26]. Such normalization of CRP values, however, may not indicate bacterial eradication but may simply reflect the fact that the inflammatory response to osteomyelitis is too localized or weak to trigger CRP production from hepatocytes [18]. In fact, the CRP level for 79.4% of our study population returned to less than 5 mg/dL in the second week after PVO diagnosis (Group 2, S2A Fig). Similar clinical observations were reported by Michail and associates, who found that all studied inflammatory markers except ESR returned to a near-normal range after 3 weeks of therapy and suggested that ESR be used to follow-up patients with osteomyelitis [16].

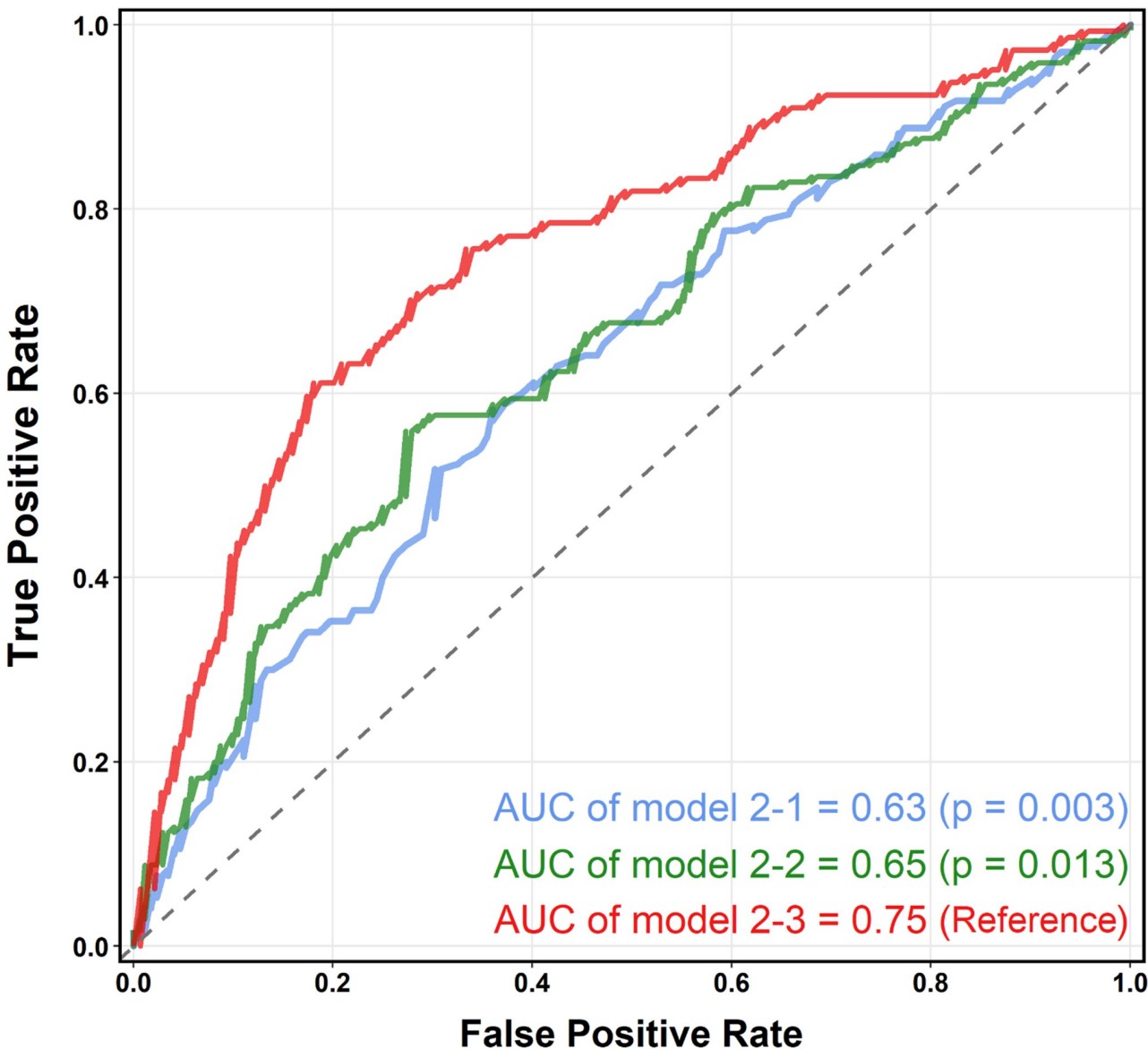

**Fig 4. Receiver operating characteristic curve for the predictive model of treatment duration ≥12 weeks. Footnotes:** Predictors in Model 2–1: Initial ESR, ESR − AD ≥ −9. Predictors in Model 2–2: Initial ESR, ESR − AD ≥ −9, ESR − CV. Predictors in Model 2–3: Initial ESR, ESR − AD ≥ −9, ESR − CV, initial CRP, age, gender, diabetes, ESRD, malignancy, CCI ≥ 3, abscess, delayed operation. **Abbreviations:** AD, absolute difference; AUC, area under the ROC curve; CCI, Charlson's comorbidity index; CRP, C-reactive protein; CV, coefficient of variation; ESR, erythrocyte sedimentation rate; ESRD, end-stage renal disease; PVO, pyogenic vertebral osteomyelitis; ROC, receiver operating characteristic curve.

Second, through GBTM, we identified three unique ESR trajectory patterns within the first 6 months following PVO diagnosis that were well characterized by clinical features, treatment duration, and recurrence rate among patients with PVO. Few studies have explored the long-term serial ESR changes (ESR trajectory) among patients with osteomyelitis [21, 22, 24]. Two studies have collapsed the serial ESR information into a single average ESR value over the course of osteomyelitis [22, 24]. Because the therapeutic response may vary throughout the

**Table 5. Prediction of 6-month recurrence using logistic regression model based on erythrocyte sedimentation rate within 4 weeks of pyogenic vertebral osteomyelitis diagnosis.**

| Variables | Crude OR (95% CI) | Model 3–1 (AIC: 253.3) [b] | | Model 3–2 (AIC: 253.8) [c] | | Model 3–3 (AIC: 220.6) [d] | |
|---|---|---|---|---|---|---|---|
| | | β | Adjusted OR (95% CI) | β | Adjusted OR (95% CI) | β | Adjusted OR (95% CI) |
| Initial ESR, mm/h | 1.01 (1.00, 1.02) | 0.009 | 1.01 (1.00, 1.02) | 0.012 | 1.01 (1, 1.03) | 0.007 | 1.01 (0.99, 1.02) |
| ESR–AD $\geq$ −9 mm/h [a] | 0.88 (0.46, 1.69) | 0.041 | 1.04 (0.52, 2.08) | 0.266 | 1.31 (0.6, 2.89) | 0.061 | 1.06 (0.41, 2.81) |
| ESR–CV | 1.01 (0.99, 1.02) | | | 0.013 | 1.01 (0.99, 1.03) | 0.016 | 1.02 (0.99, 1.04) |
| Initial CRP, mg/dL | 1.04 (1.00, 1.07) | | | | | 0.034 | 1.04 (0.99, 1.08) |
| Age, year | 1.00 (0.98, 1.02) | | | | | -0.004 | 1.00 (0.96, 1.03) |
| Male | 1.60 (0.85, 3.17) | | | | | 0.362 | 1.44 (0.62, 3.55) |
| Diabetes mellitus | 1.19 (0.64, 2.15) | | | | | -0.080 | 0.92 (0.40, 2.05) |
| ESRD | 0.75 (0.17, 2.24) | | | | | -1.986 | 0.14 (0.01, 1.08) |
| Malignancy | 1.59 (0.57, 3.87) | | | | | 0.638 | 1.89 (0.48, 6.62) |
| Any abscess | 0.94 (0.47, 2.02) | | | | | 0.438 | 1.55 (0.53, 5.74) |
| Delayed operation | 1.04 (0.45, 2.17) | | | | | -0.278 | 0.76 (0.21, 2.2) |
| eGFR, mL/min/1.73 m$^2$ | 1.00 (0.99, 1.01) | | | | | -0.008 | 0.99 (0.98, 1.01) |
| CCI $\geq$ 3 | 1.68 (0.86, 3.17) | | | | | 0.652 | 1.92 (0.68, 5.15) |
| *C* statistic (95% CI) | | 0.56 (0.46–0.65) | | 0.56 (0.46–0.66) | | 0.69 (0.61–0.78) | |

**Abbreviations:** AD, absolute difference; AIC, Akaike Information Criterion; CCI, Charlson's comorbidities index; CI, confidence interval; CRP, C-reactive protein; CV, coefficient of variation; eGFR, estimated glomerular filtration rate; ESR, erythrocyte sedimentation rate; ESRD, end-stage renal disease; OR, odds ratio; PVO, pyogenic vertebral osteomyelitis.

[a]. We used the median value of ESR–AD as the cutoff value.

[b]. In Model 3–1, initial ESR, ESR–AD were included as predicting variables and 6-month recurrence as the dichotomous response variable.

[c]. In Model 3–2, initial ESR, ESR–AD, and ESR–CV were included as predicting variables and 6-month recurrence as the dichotomous response variable.

[d]. In Model 3–3 (full model), initial ESR, ESR–AD, ESR–CV, initial CRP, demographic information (age, gender), baseline comorbidities (diabetes, ESRD, malignancy, and CCI), presence of abscess, delayed operation, and baseline eGFR were included as predicting variables and 6-month recurrence as the dichotomous response variable.

disease course and between individuals, this approach would overlook the inter-patient heterogeneity in treatment response and is unable to characterize the prognostic role of ESR [13]. Contrarily, GBTM assumes that the population is composed of distinct groups, each with a different underlying trajectory [27]. As demonstrated in our study, the three ESR trajectories represented distinct PVO phenotypes exhibiting significantly different treatment durations and recurrence rates. To the best of our knowledge, it is the first time this approach was used to classify and predict the clinical course of PVO.

Third, the timing of surgical intervention for PVO remains controversial [3]. Segreto et al found an immediate operation defined by within 24 hours of admission may reduce complications and mortality among patients with PVO in a claim database [28]. Other studies based on single-center experience showed no significant benefit from early abscess drainage [29, 30], which was consistent to our findings. The challenges in tackling the issues related to the timing of surgical intervention, researchers must ensure the timing of PVO diagnosis can be standardized. However, the prerequisite condition is hardly satisfied, if not impossible. Therefore, immediate antibiotics treatment upon the diagnosis of PVO with duration guided by ESR response is still the treatment of choice for PVO. Clinicians should pay attention to the surgical indication, but the timing may be individualized according to patient's response and symptoms.

Fourth, the proposed models are able to predict the longitudinal ESR trajectories and clinical course of PVO. Two studies have suggested that specific values of ESR and CRP during

treatment predict osteomyelitis recurrence [4, 23]; however, this prediction may be misleading because of potential reverse causation. By contrast, we proposed to use the initial ESR and the first-4-week ESR variability to predict the clinical course of PVO. Despite the weak predicative performance and moderate-to-good discrimination−$C$ statistic ranging from 0.63 (two variables) to 0.75 (full model) for long-term treatment ($\geq$12 weeks) and from 0.56 (two variables) to 0.69 (full model) for recurrence−it is the first evidence that verifies the predictive role of ESR in a real-world setting. From a clinical point of view, the predictive probability graph (Fig 3) using simple parameters (initial ESR and dichotomous ESR-AD) can help physicians assess earlier whether patients with PVO would have slow response in ESR and necessitate a prolonged antibiotic treatment or have a high likelihood of recurrence. From the $C$ statistic results from the simple model to the full model, we realized that clinical information other than ESR also influences the treatment duration and recurrence rate of PVO.

This study has several limitations. First, this was a retrospective study, and the time points and frequencies of ESR measurements following PVO diagnosis were not standardized; variations in these measurements might have affected the ESR trajectory patterns. However, the total number of ESR measurements in this study was 3524, which minimize the confounding by sampling indication. Moreover, the median time intervals of sampling were similar within the first 4 weeks of PVO across the three ESR trajectory groups. Second, we excluded patients who had fewer than two ESR measurements within 6 months following PVO diagnosis. Such exclusion might cause selection bias and influence the ESR trajectory pattern. Only 13 out of 414 cases (3%) were excluded from this study; therefore, the possibility of selection bias was minimal. Third, residual confounding, in particular regarding nutritional status and immune function, was not factored into the proposed models. The dynamics of ESR, however, might serve as an overall summation of all the host–pathogen elements in play. Finally, this study was performed at a single site, and this may limit its generalizability. Nonetheless, from a pathophysiological perspective, our findings can be reliably applied to adult patients with PVO.

In conclusion, the results of this study demonstrate that the PVO population is composed of distinct groups based on ESR trajectory. In addition, the clinical characteristics, treatment duration, and recurrence rate among the patients with PVO are linked with particular ESR trajectory patterns. Using the models proposed in this study and a predictive probability graph, clinicians can predict particular ESR trajectories for patients with PVO and determine which patients require prolonged treatment duration or have a high risk for PVO recurrence. Our findings provide new perspectives within the study of PVO but need to be validated in the future.

## Supporting information

**S1 Table. Variability of erythrocyte sedimentation rate during the first 4 weeks following pyogenic vertebral osteomyelitis diagnosis ($N$ = 342), based on treatment duration and recurrence status.**
(DOCX)

**S2 Table. The 4-week erythrocyte sedimentation rate (ESR) variability and 6-month ESR trajectory by the surgical treatment.**
(DOCX)

**S3 Table. The 4-week erythrocyte sedimentation rate (ESR) variability and 6-month ESR trajectory by the presence of abscess.**
(DOCX)

**S4 Table. Prediction of erythrocyte sedimentation rate (ESR) trajectory group (Groups 1 vs. 2+3) through logistic regression based on ESR within 4 weeks of pyogenic vertebral osteomyelitis diagnosis.**
(DOCX)

**S1 Fig. Directed acyclic graph (DAG) of the potential causal relationship between covariables and poor prognosis outcomes among patients with PVO.**
(DOCX)

**S2 Fig. Group-based trajectory modeling of 6-month C-reactive protein (CRP) and 6-month white blood cell count (WBC) among patients with pyogenic vertebral osteomyelitis.**
(DOCX)

**S3 Fig. Receiver operating characteristic curve for the predictive model of 6-month recurrence.**
(DOCX)

**S4 Fig. Calibration plot for the full predictive model of treatment duration $\geq$12 weeks (model 2–3).**
(DOCX)

**S5 Fig. Calibration plot for the full predictive model of 6-month recurrence (model 3–3).**
(DOCX)

## Author Contributions

**Conceptualization:** Hsiu-Yin Chiang, Chih-Yu Chi.

**Data curation:** Hsiu-Yin Chiang, Wei-Shuo Chang, Chih-Yu Chi.

**Formal analysis:** Hsiu-Yin Chiang, Chih-Wei Chung, Yen-Chun Lo.

**Methodology:** Hsiu-Yin Chiang, Chin-Chi Kuo.

**Project administration:** Chin-Chi Kuo.

**Software:** Chih-Wei Chung, Yen-Chun Lo.

**Supervision:** Chin-Chi Kuo.

**Validation:** Chin-Chi Kuo.

**Writing – original draft:** Hsiu-Yin Chiang, Chin-Chi Kuo, Chih-Yu Chi.

**Writing – review & editing:** Chin-Chi Kuo.

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
