## [Decision Letter · Decision Letter 0]

2 Sep 2019

PONE-D-19-15739

First-4-Week Erythrocyte Sedimentation Rate Variability Predicts Erythrocyte Sedimentation Rate Trajectories and Clinical Course among Patients with Pyogenic Vertebral Osteomyelitis

PLOS ONE

Dear Dr. Kuo,

Thank you for submitting your manuscript to PLOS ONE. After careful consideration, we feel that it has merit but does not fully meet PLOS ONE’s publication criteria as it currently stands. Therefore, we invite you to submit a revised version of the manuscript that addresses the points raised during the review process.

We would appreciate receiving your revised manuscript by Oct 17 2019 11:59PM. To enhance the reproducibility of your results, we recommend that if applicable you deposit your laboratory protocols in protocols.io, where a protocol can be assigned its own identifier (DOI) such that it can be cited independently in the future. For instructions see: http://journals.plos.org/plosone/s/submission-guidelines#loc-laboratory-protocols

We look forward to receiving your revised manuscript.

Kind regards,

Daniel Pérez-Prieto, PhD

Academic Editor

PLOS ONE

Journal Requirements:

1. Please include your tables as part of your main manuscript and remove the individual files. Please note that supplementary tables (should remain/ be uploaded) as separate "supporting information" files

Reviewers' comments:

Reviewer's Responses to Questions

**Comments to the Author**

1. Is the manuscript technically sound, and do the data support the conclusions?

Reviewer #1: Yes

Reviewer #2: Yes

2. Has the statistical analysis been performed appropriately and rigorously? 

Reviewer #1: Yes

Reviewer #2: I Don't Know

3. Have the authors made all data underlying the findings in their manuscript fully available?

Reviewer #1: Yes

Reviewer #2: Yes

4. Is the manuscript presented in an intelligible fashion and written in standard English?

Reviewer #1: Yes

Reviewer #2: Yes

5. Review Comments to the Author

Reviewer #1: I would suggest minor revions/ clarifications to the manuscript.

- line 77 highest ESR?

- What was the rational for considering slow vs. fast decay? Considering the figure 2, what was the explanation for the slope in the group 3 (initial high, slow decay) at week 6, as being coincident with the group 2 (initial high, fast decay)?

- Clarification of the mixed/synchronous infections. If there are patients with complex infections, as for example concomitant Endocarditis they should be excluded, as it is expected to have higher ESR and persist more than >4 weeks, which could led to workup bias.

- Would be interesting more details on the timing of the delayed surgery, at week 2? At recurrence? and include at the discussion. The authors should add a comment on the presence of abscess without surgery - 18-20% in each group, and the impact of ESR.

- line 309 concluding that the assessment of prolong antimicrobial therapy based on ESR is an extrapolation of data analysis, and gets to the previous point, as further analysis of the delayed or no surgery vs. duration of therapy or recurrence needed.

Reviewer #2: The authors present a retrospective observational cohort study, which includes a high number of cases of vertebral osteomyelitis (more than 400) from a single center. The study exhaustively analyzes the relationship between the evolution of ESR in the first 4 weeks and the evolution of vertebral osteomyelitis, in terms of duration of antibiotic treatment, recurrence and mortality. They find a relationship between the evolution of ESR levels and the duration of antibiotic therapy and recurrences.

Abstract:

- The authors define the existence of 3 trajectory pattern (of ESR evolution), but only 1 is described in the abstract (which is the one that has an influence on the increased risk of recurrence. briefly define the 3 trajectory pattern); some brief information should be done of the 3 patterns.

Methods:

- If there is no space restriction, it would be advisable to explain the study inclusion and exclusion criteria and definitions of the variables; this would be more recommendable In case of space problem, the most important criteria and variables should be briefly explained.

- One of the most relevant outcomes in this study is recurrence. In a retrospective study like this, the “objective” definition of the outcomes is difficult. However, the recommendation is to use a definition that is as understandable and "reproducible" as possible. In this case, "recurrence" is defined too imprecisely (e.g., according to this definition, presenting back pain within 6 months after the suspension of antibiotic treatment would be sufficient to diagnose recurrence). Actually, the definition of recurrence seems to be based on the consideration of the responsible (taking into account clinic symptoms and analytics), is that so? But, for instance, were some cases considered recurrence by the attending physician excluded of this outcome due to lack of data supporting this diagnosis?

- “if a surgical intervention was performed> 2 weeks after PVO diagnosis, it was defined as a delayed operation” Please, could you explain the meaning of this sentence in more detail?. I guess do you refer to a surgical intervention to treat the infection, although it could also refer to corrective surgery (by sequelae)…. If it is a surgical treatment of the infection, does it refer to patients who had an initial indication of surgery, but was it performed “late”? Or who were operated late for lack of initial clinical and / or radiological improvement with antibiotic treatment (but without apparent initial surgical indication? The interpretation of the surgical variable can be very different (“consequence” of bad evolution or “cause” of bad evolution due to the non-early performance)

- The statistical analysis does not specifically mention the performance of trend analysis, which is after that cited in several sections.

Results:

- The reference to Figure 1 should be at the end of line 176

- Figure 2: it is not necessary to repeat the number (1, 2 or 3) on each line

- Line 193: “P for trend” à the meaning is not explained neither has been previously mentioned the use of this statistics; so, it should be clarified

- Table 1: the meaning of “NA” is not defined. In some cases there is a result of "p for trend" and in others apparently not (that is, in these cases the "p" would only reflect that there is some difference between the 3 groups?); this issue should be clarified

- It would be interesting to know if group 3 had overall more comorbidities (or any comorbidity) compared to the other groups, apart from the specific comorbidities mentioned (lines 192-194)

- Were there patients with a simultaneous diagnosis of endocarditis? Was this possibility assessed regularly among patients with infection or bacteraemia documented by gram-positive cocci?

- Could it be analyzed if the indication for immediate surgical treatment changed over time? A percentage of 50% seems very high for current practice

- The 3 defined trajectory pattern show evident differences in the starting point and evolution of the ESR. However, finally, the prognostic differences seem to be found only between two groups (3 versus 1 + 2 or 1 with respect to 2 + 3 groups). The text and tables 3-5 do not quite clarify this point. In this situation, perhaps after a first descriptive analysis of the 3 groups, the most sophisticated (multivariate) analysis of the outcomes could be done considering the 2 groups that present prognostic differences in the analyzed outcomes; this, in addition, would increase the statistical power to be able to detect differences.

- In tables 3-5, the different models performed should be explained at the bottom of the table (the tables must be self-explanatory).

- In the multivariate models described in Tables 3-5, how have been decided which variables to include? It seems somewhat arbitrary ... Have all the variables that could influence the prognosis been considered (particularly those kwown variables increasing the risk of recurrence?

- I suppose that it has been decided to analyze the ESR because it presents a better predictive profile of the evolution of vertebral osteomyelitis. However, I would like to know what the situation in the case of the ERP and why the analysis of this marker was ruled out. In principle, it is thought that ERP could be the parameter of choice. The evolution of leukocytes, ERP and ESR in a single graph could be of interest; it could also be considered to analyze the mentioned markers according to the current 3 defined trajectory patterns. In supplementary figure 1, it is clear that the division into 3 groups of patients in relation to CPR is not useful; in this sense, surgical treatment should probably have an important role that makes interpretation difficult in these patients (in which CPR should have a significant increase in relation to surgery that distorts the overall results); maybe the ERP analysis could be considered excluding surgery patients or excluding ERP increases related to surgery.

- Figure 3 is not very intuitive

Discussion

- Lines 275: “The inflammatory marker CRP returns to its normal level within 1 week”. Maybe it means "may" (or something similar) returns?

6. PLOS authors have the option to publish the peer review history of their article (what does this mean?). If published, this will include your full peer review and any attached files.

Reviewer #1: No

Reviewer #2: No

---

## [Author Response · Author response to Decision Letter 0]

19 Oct 2019

Author Response

First-4-Week Erythrocyte Sedimentation Rate Variability Predicts Erythrocyte Sedimentation Rate Trajectories and Clinical Course among Patients with Pyogenic Vertebral Osteomyelitis (PONE-D-19-15739)

Review Comments to the Author

Reviewer #1: 

1. I would suggest minor revions/ clarifications to the manuscript.

Response: Thank you for the comments. We have addressed each of the Reviewers’ comments as follows.

Revised contents: None.

2. line 77 highest ESR?

Response: We have carefully read the study done by Lin et al. (Reference 4 in the revised Text) and confirmed this should be the “lowest” ESR. Lin et al. compared the predictive performance of the different cut-off points (20, 25, 30 mm/hr) for the lowest ESR in predicting recurrence [1]. They identified that the cut-off of >=20 mm/hr for the lowest ESR had the best sensitivity (85%) in predicting recurrence and used that cut-off in their subsequent analysis.

Revised contents: None.

3. What was the rational for considering slow vs. fast decay? Considering the figure 2, what was the explanation for the slope in the group 3 (initial high, slow decay) at week 6, as being coincident with the group 2 (initial high, fast decay)?

Response: We thank Reviewer 1 for the opportunity to clarify and improve our study. 

1) Using the group-based trajectory statistical modeling (GBTM) on all ESR values measured throughout the first 6 months following PVO diagnosis, we were able to identify three distinct ESR trajectories: Group 1 (initial moderate, fast decay), Group 2 (initial high, fast decay), and Group 3 (initial high, slow decay). In clinical interpretation, slow-decay ESR represents a slow recovery or response of ESR to treatment and fast-decay ESR represents a prompt recovery or response of ESR to treatment. Therefore, we replaced the term “decay” with “response” throughout the manuscript in order to bring more clinical meaning to readers. 

2) We thank Reviewer’s observation that Group 3’s (initial-high slow-response) mean ESR dropped suddenly at week 6, which coincided with Group 2’s mean ESR at week 6. The distribution of ESR for week 5, 6, and 7 are listed in Table R1 below. We confirmed that the actual mean value of ESR for Group 2 and Group 3 indeed decreased dramatically from week 5 to week 6 and the two mean values at week 6 are similar (Group 2 vs 3: 46.59 vs 49.11 mm/hr). This could be well explained by the method we used to model longitudinal trajectories of ESR, a.k.a., Group-Based Trajectory Modeling (GBTM). 

Statistically, the unexpected low mean ESR at week 6 for Group 3 is caused by the skewed ESR distribution at week 6. If the distribution of a measure is skewed (i.e., not normally distributed), the mean can be easily affected by the extremely large or extremely small value. For Group 3, at least 25% of patients who had ESR measured at week 6 had an ESR of 11 mm/hr or less and the median was larger than the mean (56 vs 49.11 mm/hr). Therefore, the ESR distribution at week 6 is skewed to the left and median (56 mm/hr) can better represent the overall measure here. Furthermore, GBTM is a specialized application of finite mixture modeling designed to identify clusters of individuals who follow similar trajectories. Therefore, trajectories are what we really focus on, not the change at a single timepoint.

Table R1. ESR distribution at week 5, 6, and 7 following the PVO diagnosis.

　 Week Available N Minimal 25th percentile Median Mean 75th percentile Maximum

Group 1

(N = 122) Week 5 64 3.00 21.75 28.50 30.48 38.00 86.00

 Week 6 48 8.00 18.00 24.50 28.77 38.50 67.00

 Week 7 44 3.00 14.50 21.00 25.16 33.25 67.00

Group 2

(N = 144) Week 5 75 15.00 51.75 67.00 67.85 83.00 122.00

 Week 6 70 10.00 33.25 49.50 46.59 62.75 90.00

 Week 7 71 16.00 40.50 57.00 58.61 75.00 113.00

Group 3

(N = 135) Week 5 76 37.00 77.50 91.50 89.61 106.00 129.00

 Week 6 70 10.00 11.00 56.00 49.11 82.75 99.00

 Week 7 65 15.00 67.00 87.00 85.31 102.00 130.00

Revised content: We replaced the term “decay” with “response” throughout the manuscript.

4. Clarification of the mixed/synchronous infections. If there are patients with complex infections, as for example concomitant Endocarditis they should be excluded, as it is expected to have higher ESR and persist more than >4 weeks, which could led to workup bias.

Response:

1) We thank Reviewer’s suggestions that allow us to clarify our study. Mixed and synchronous infections have distinct definitions in our study. Mixed infections are bone infections that were caused by more than one bacterial species. We have also replaced the term “mixed” with “polymicrobial” to improve the clarity. Synchronous infections are the concurrent infections at body sites that are distant from the primary vertebral nidus such as lung (pneumonia), urinary tract, intra-abdomen, bloodstream, or skin and soft tissues. We have added the description for mixed and synchronous infections in MATERIALS and METHODS section.

2) Of 401 patients in the study population, we identified only 3 patients (0.75%) had infective endocarditis (IE) that met the modified Duke criteria within 30 days prior to the PVO diagnosis and all of them had been treated adequately. In addition, only 1 of these 3 patients were classified in the ESR Group 3 that had high initial ESR and responded slow. Thus, we believe that work-up bias and information bias are unlikely and excluding 3 patients with IE would not change our findings.

Revised content:

1) Page 6, Line 123: In MATERIALS and METHODS section under Covariables and outcomes, we added definition of mixed infection and synchronous infections as follows: “The covariables included the comorbidities of the patients within 1 year prior to or at the time of PVO diagnosis (e.g., diabetes mellitus, intravenous drug users, liver cirrhosis, end stage renal disease, malignancy, other comorbidities, and Charlson’s comorbidity index [CCI], etc.), culture results (i.e., gram-negative pathogens, gram-positive pathogens, polymicrobial microorganisms [≥ 2 different pathogens], or no growth), pre-existing or synchronous infections from distant body sites such as lung (pneumonia), urinary tract, intra-abdomen, bloodstream, or skin and soft tissues within 1 month prior to or at the time of PVO diagnosis, timing of surgical intervention, and biochemical profiles.”

2) Page 7, Line 127; Page 9, Line 192; Page 11, Line 225; Page 13, Line 271: We replaced the term “mixed infection/culture” with “polymicrobial infection/culture”.

3) Table 1 & Table 3: We replaced the term “mixed” with “polymicrobial”.

5. Would be interesting more details on the timing of the delayed surgery, at week 2? At recurrence? and include at the discussion. The authors should add a comment on the presence of abscess without surgery - 18-20% in each group, and the impact of ESR.

Response: 

1) We thank Reviewer 1’s valuable comments. In the present study, two Infectious Disease Specialists (CCY and WSC) had reviewed the medical records for all patients and defined patients receiving delayed surgery if they had initial indications for surgical intervention, such as neural compression, spinal instability, or presence of epidural or paravertebral abscesses, at the time of PVO diagnosis but operated (e.g., discectomy, laminectomy) ≥ 2 weeks after the diagnosis of PVO [2].

2) We agree with Reviewer 1 that the delayed operation and the presence of abscess could influence patients’ ESR levels. In Table 1 in the main text, delayed operation was more common in ESR Group 2 (initial-high, fast-response; 18.8%) and Group 3 (initial-high, slow-response; 21.5%), compared with that in Group 1 (initial-moderate, fast-response; 7.4%). In addition, the initial ESR was slightly higher (mean: 80.6 vs 75.8 mm/hr; P = 0.196) and the absolute decrease in ESR within the first 4 weeks was less (mean: -4.4 vs -9.3 mm/hr; P = 0.269) in patients receiving delayed operations than in patients receiving immediate operations (Table R2). Furthermore, 86.2% of patients with delayed operations were classified as ESR Group 2 or Group 3, compared with 68.9% of patients with immediate operations. Thus, patients with delayed operations tended to have high initial ESR and slowly responded ESR level, compared with those with immediate operations. However, the treatment duration (delayed vs immediate operation, 118 vs 100 days; P = 0.097) and the 6-month recurrence (14.5% vs 15.8%; P =0.810) were similar.

 

Table R2. The 4-week ESR variability and 6-month ESR trajectory by the surgical treatment.

 Immediate operation

(N = 206) Delayed operation

(N = 65) Drainage without operation

(N = 21) No operation

(N = 109) p-value

ESR variability within 4 weeks of PVO diagnosis, mean ± SD 

Initial ESR 75.8 ± 29.5 b 80.6 ± 26.8 b 67.4 ± 18.3 74.5 ± 27.2 0.587

ESR-AD -9.3 ± 29.9 c -4.4 ± 30.9 c -14.1 ± 33.4 -11.6 ± 25.9 0.489

ESR-CV 26.1 ± 18.1 20.8 ± 14.1 27.6 ± 20.9 23.2 ± 18.0 0.254

ESR-PC 9.5 ± 202.6 3.7 ± 49.2 -13.0 ± 48.1 -9.6 ± 49.0 0.315

ESR-intercept 76.3 ± 21.1 83.5 ± 20.3 69.8 ± 12.1 77.4 ± 20.3 0.845

ESR-Slope -0.4 ± 0.7 -0.4 ± 0.7 -0.7 ± 0.9 -0.5 ± 0.6 0.220

ESR trajectory within 6 months of PVO diagnosis a, n (%) 0.054

Group 1 64 (31.1) 9 (13.9) 8 (38.1) 41 (37.6) 

Group 2 74 (35.9) 27 (41.5) 6 (28.6) 37 (33.9) 

Group 3 68 (33.0) 29 (44.6) 7 (33.3) 31 (28.5) 

Treatment duration, day, mean ± SD 100.4 ± 77.0 d 118.4 ± 89.8 d 80.2 ± 51.9 111.2 ± 76.4 0.375

Recurrence, n (%) 30 (15.8) e 9 (14.5) e 5 (25.0) 9 (8.7) 0.149

Abbreviations: AD, absolute difference; CV, coefficient of variation; ESR, erythrocyte sedimentation rate; PC, percent change; PVO, pyogenic vertebral osteomyelitis; SD, standard deviation.

a. Group 1: initial-moderate, fast-response; Group 2: initial-high, fast-response; Group 3: initial-high, slow-response.

b. The initial ESR for patients with immediate operation and those with delayed operation was not significantly different (P = 0.196).

c. The ESR-AD for patients with immediate operation and those with delayed operation was not significantly different (P = 0.269).

d. The treatment duration for patients with immediate operation and those with delayed operation was not significantly different (P = 0.097).

e. The recurrence status for patients with immediate operation and those with delayed operation was not significantly different (P = 0.810).

3) As shown in Table 1 in the main text, the presence of abscess without drainage/operation was comparable across the three ESR trajectory groups (Group 1: 20.5%; Group 2: 18.1%; Group 3: 17.8%). The initial ESR was slightly lower for patients without abscess than those with abscess (no abscess: 70.5; abscess without drain: 77.3; abscess with drain: 76.8 mm/hr; Table R3), but was comparable between abscess with and without drainage/operation (P = 0.918). The proportion of PVO patients receiving abscess drainage that was classified into ESR Group 2 or Group 3 were also comparable (Table R3). Thus, patients with abscess tended to have higher initial ESR and slowly responded ESR level, but having drainage/operation or not did not affect their ESR patterns. However, among patients with PVO accompanied with paraspinal abscesses, drainage itself did not alter the disease course in the regard of treatment duration (no drainage vs drainage, 116 vs 103 days; P = 0.077) or the 6-month recurrence (6.8% vs 16.2%; P = 0.044). 

Table R3. The 4-week ESR variability and 6-month ESR trajectory by the presence of abscess.

Variables No abscess 

(N = 80) Abscess without drainage/ operation 

(N = 75) Abscess 

with

drainage/ operation 

(N = 246) p-value

ESR variability within 4 weeks of PVO diagnosis, mean ± SD 

Initial ESR 70.5 ± 29.2 77.3 ± 26.2 b 76.8 ± 28.3 b 0.181

ESR-AD -11.1 ± 30.5 -11.4 ± 25.0 -8.3 ± 30.1 0.418

ESR-CV 25.3 ± 14.8 22.5 ± 19.1 25.1 ± 18.1 0.810

ESR-PC -5.9 ± 55.4 -9.7 ± 49.1 8.9 ± 189.9 0.407

ESR-intercept 73.9 ± 20.4 79.8 ± 19.9 77.5 ± 20.8 0.382

ESR-Slope -0.5 ± 0.7 -0.5 ± 0.6 -0.4 ± 0.7 0.199

ESR trajectory within 6 months of PVO diagnosis a, n (%) 0.114

Group 1 33 (41.2) 25 (33.3) 64 (26.0) 

Group 2 22 (27.5) 26 (34.7) 96 (39.0) 

Group 3 25 (31.2) 24 (32.0) 86 (35.0) 

Treatment duration, day, mean ± SD 100.1 ± 92.7 116.3 ± 72.4 c 103.5 ± 74.8 c 0.986

Recurrence, n (%) 11 (14.7) 5 (6.8) d 37 (16.2) d 0.133

Abbreviations: AD, absolute difference; CV, coefficient of variation; ESR, erythrocyte sedimentation rate; PC, percent change; PVO, pyogenic vertebral osteomyelitis; SD, standard deviation.

a. Group 1: initial-moderate, fast-response; Group 2: initial-high, fast-response; Group 3: initial-high, slow-response.

b. The initial ESR for patients who had abscess without drainage and those who had abscess with drainage was not significantly different (P = 0.918).

c. The treatment duration for patients who had abscess without drainage and those who had abscess with drainage was not significantly different (P = 0.077).

d. The recurrence status for patients who had abscess without drainage and those who had abscess with drainage was significantly different (P = 0.04).

In our original multivariable analyses, we have adjusted for delayed operation and the presence of abscess in the analysis for prolonged treatment duration (Table 4 in the main text) and recurrence (Table 5 in the main text). Neither delayed operation (adjusted OR, 1.49; 95% CI, 0.71 – 3.22) nor the presence of abscess (adjusted OR, 1.97; 95% CI, 0.98 – 4.04) was significantly associated with prolonged treatment. Similarly, no significant association between delayed operation or abscess with recurrence was observed. 

Therefore, we have included delayed operation, presence of abscess, and ESR variability measures (initial ESR, ESR-AD, and ESR-CV) in the fully adjusted model, which have adjusted for the potential confounding induced by delayed operation and presence of abscess on outcomes.

Revised content:

1) Page 14, Line 283: In RESULTS section, we added the following: “Since the delayed operation and the presence of abscess could influence patients’ ESR levels and confound the association between ESR level and outcome (Supplementary Table 2 and 3), we included these variables along with the clinical information, such as the comorbidities, delayed surgery, and abscess formation in the full model.”

2) Page 16, Line 339: In DISCUSSION section, we added the third discussion point: “Third, the timing of surgical intervention for PVO remains controversial [3]. Segreto et al found an immediate operation defined by within 24 hours of admission may reduce complications and mortality among patients with PVO in a claim database [4]. Other studies based on single-center experience showed no significant benefit from early abscess drainage [5, 6], which was consistent to our findings. The challenges in tackling the issues related to the timing of surgical intervention, researchers must ensure the timing of PVO diagnosis can be standardized. However, the prerequisite condition is hardly satisfied, if not impossible. Therefore, immediate antibiotics treatment upon the diagnosis of PVO with duration guided by ESR response is still the treatment of choice for PVO. Clinicians should pay attention to the surgical indication, but the timing may be individualized according to patient’s response and symptoms.”

3) Supplementary: We added Supplementary Table 2 and Table 3 to show the ESR variability and ESR trajectory grouping by the surgical treatment or by the presence of abscess.

6. Line 309 concluding that the assessment of prolong antimicrobial therapy based on ESR is an extrapolation of data analysis, and gets to the previous point, as further analysis of the delayed or no surgery vs. duration of therapy or recurrence needed.

Response:

We thank Reviewer 1’s suggestion. We agree that delayed operation could influence the ESR pattern and contribute to the prolonged treatment duration or recurrence (aka: an important confounding factor). However, the predictive probability estimated based on a practical model is not an extrapolation, but a practical conversion from statistical modeling into clinical use. We have examined the 4-week ESR variability and the 6-month ESR trajectory by the operation timing (i.e., immediate operation, delayed operation, drainage only, no operation) in Table R2. Patients with delayed operations tended to have high initial ESR and slowly responded ESR level, compared with those with immediate operations. However, the treatment duration and the 6-month recurrence were not significantly different (Table R2). Please also see our response to the comment #5 for more details.

Revised content:

1) Page 14, Line 283: In RESULTS section, we added the following: “Since the delayed operation and the presence of abscess could impact patients’ ESR levels and confound the association between ESR level and outcome (Supplementary Table 2 and 3), we included these variables along with the clinical information, such as the comorbidities, delayed surgery, and abscess formation in the full model.”

2) Page 16, Line 339: In DISCUSSION section, we added the third discussion point: “Third, the timing of surgical intervention for PVO remains controversial [3]. Segreto et al found an immediate operation defined by within 24 hours of admission may reduce complications and mortality among patients with PVO in a claim database [4]. Other studies based on single-center experience showed no significant benefit from early abscess drainage [5, 6], which was consistent to our findings. The challenges in tackling the issues related to the timing of surgical intervention, researchers must ensure the timing of PVO diagnosis can be standardized. However, the prerequisite condition is hardly satisfied, if not impossible. Therefore, immediate antibiotics treatment upon the diagnosis of PVO with duration guided by ESR response is still the treatment of choice for PVO. Clinicians should pay attention to the surgical indication, but the timing may be individualized according to patient’s response and symptoms.”

3) Page 17, Line 361: We modified the following sentence: “From a clinical point of view, the predictive probability graph (Figure 3) using simple parameters (initial ESR and dichotomous ESR-AD) can help physicians assess earlier whether patients with PVO would have slow response in ESR and necessitate a prolonged antibiotic treatment or have a high likelihood of recurrence.”

4) Supplementary: We added Supplementary Table 2 and Table 3 to show the ESR variability and ESR trajectory grouping by the surgical treatment or by the presence of abscess.

 

Reviewer #2: 

1. The authors present a retrospective observational cohort study, which includes a high number of cases of vertebral osteomyelitis (more than 400) from a single center. The study exhaustively analyzes the relationship between the evolution of ESR in the first 4 weeks and the evolution of vertebral osteomyelitis, in terms of duration of antibiotic treatment, recurrence and mortality. They find a relationship between the evolution of ESR levels and the duration of antibiotic therapy and recurrences.

Response: We are very grateful for the reviewer’s positive comments/suggestions and for the insightful reviews. Below we describe our responses point-by-point to each comment given in italics. In the revised manuscript, changes to the text are noted with red text. 

Revised content: None.

Abstract:

2. The authors define the existence of 3 trajectory pattern (of ESR evolution), but only 1 is described in the abstract (which is the one that has an influence on the increased risk of recurrence. Briefly define the 3 trajectory pattern); some brief information should be done of the 3 patterns.

Response: Thank you. We have added a brief description of the 3 trajectory patterns in the abstract. 

Revised content:

Page 2, Line 50: In ABSTRACT section under Results, we added: “Three ESR trajectory patterns were identified though GBTM among patients with PVO: Group 1, initial moderate high ESR with fast response; Group 2, initial high ESR with fast response; Group 3, initial high ESR with slow response.”

 

Methods:

3. If there is no space restriction, it would be advisable to explain the study inclusion and exclusion criteria and definitions of the variables; this would be more recommendable In case of space problem, the most important criteria and variables should be briefly explained.

Response: We thank Reviewer 2’s valuable suggestions. We previously did not give much details about the inclusion/exclusion criteria because our study population has been described in our prior published work [2] However, we agree with Reviewer 2 that it would be better comprehensible if we provide more details. Therefore, we have expanded the METHODS section to elaborate the inclusion/exclusion criteria and the definition of important variables in our study. 

Revised content:

1) Page 6, Line 107: In MATERIALS and METHODS section under Study design and study population, we added the inclusion and exclusion criteria of the study population as follows: “Briefly, patients were excluded if their antibiotics treatment was shorter than 14 days, if they had a non-hematogenous source of vertebral infection (i.e., artificial implants, prior laminectomy within 1 year, spine penetrating trauma, or decubitus ulcer at the same level of vertebral osteomyelitis), if their vertebral osteomyelitis were caused by mycobacterium, fungus, or Brucella species, or if they did not have any image evidence. In the current study, we further excluded patients who had fewer than two ESR measurements within 6 months following the diagnosis of PVO.”

2) Page 6, Line 123: In MATERIALS and METHODS section under Covariates and outcomes, we re-wrote as follows: “The covariables included the comorbidities of the patients within 1 year prior to or at the time of PVO diagnosis (e.g., diabetes mellitus, intravenous drug users, liver cirrhosis, end stage renal disease, malignancy, other comorbidities, and Charlson’s comorbidity index [CCI], etc.), culture results (i.e., gram-negative pathogens, gram-positive pathogens, polymicrobial microorganisms [≥ 2 different pathogens], or no growth), pre-existing or synchronous infections from distant body sites such as lung (pneumonia), urinary tract, intra-abdomen, bloodstream, or skin and soft tissues within 1 month prior to or at the time of PVO diagnosis, timing of surgical intervention, and biochemical profiles. Baseline biochemical profiles of white blood cell (WBC), c-reactive protein (CRP), ESR, blood urea nitrogen (BUN), serum creatinine (SCr), and estimated glomerular filtration rate (eGFR, calculated using chronic kidney disease epidemiology [CKD-EPI] collaboration equation) were obtained within 7 days (before or after) of PVO diagnosis. We defined surgical procedures performed within 2 weeks after PVO diagnosis as immediate operations and those performed later than that as delayed operations. Prolonged antibiotic treatment duration was defined as treatment duration longer than or equals to 12 weeks. The EMRs were reviewed by two Infectious Diseases Specialists (CYC and WSC) to determine patients’ delayed operation and recurrence status. We defined surgical procedures performed within 2 weeks after PVO diagnosis as immediate operations. Patients receiving delayed surgery were defined if they had initial indications for surgical intervention (e.g., neural compression, spinal instability, or presence of epidural or paravertebral abscesses) at the time of PVO diagnosis but were operated (e.g., discectomy, laminectomy) later than 2 weeks after the diagnosis of PVO. Recurrence was defined as any recurrent symptoms and signs (e.g., fever or pain on the affected site accompanied with abnormal image study or increased inflammatory markers, in the absence of other causes) within 6 months after the completion of the initial antibiotic treatment and received another course of antibiotic treatment based on clinician’s decision. In-hospital mortality, 3-month mortality, or 6-month mortality following the PVO diagnosis were obtained by linking to the National Death Registry of Taiwan.”

4. One of the most relevant outcomes in this study is recurrence. In a retrospective study like this, the “objective” definition of the outcomes is difficult. However, the recommendation is to use a definition that is as understandable and "reproducible" as possible. In this case, "recurrence" is defined too imprecisely (e.g., according to this definition, presenting back pain within 6 months after the suspension of antibiotic treatment would be sufficient to diagnose recurrence). Actually, the definition of recurrence seems to be based on the consideration of the responsible (taking into account clinic symptoms and analytics), is that so? But, for instance, were some cases considered recurrence by the attending physician excluded of this outcome due to lack of data supporting this diagnosis?

Response: We thank Reviewer 2 for the opportunity that allows us to clarify the definition of recurrence. The medical records were reviewed by two infectious specialists (CYC and WSC) to determine patients’ recurrence status. Recurrence was defined as having any recurrent symptoms or signs (e.g., fever, pain on the affected site, or abnormal inflammatory markers in the absence of other causes) within 6 months after the completion of the initial antibiotic treatment and received another course of antibiotic treatment based on clinician’s decision. In the clinical care for the patients with PVO after the treatment completion, clinicians often follow patients for treatment response by examining their inflammatory markers levels or MRI image study. Patients having back pain within 6 months after treatment completion would not be considered as having recurrence if they do not have clinical evidence such as increased informatory markers or abnormal image data to support. 

In our study, 48 of 53 (90.6%) patients with recurrence and 202 of 323 (62.5%) patients without recurrence had their ESR measured after the treatment completion and the days from the end of treatment to the closest ESR was comparable between two groups (median, 13 days; Table R4). The ESR value was significantly higher among patients having 6-month recurrence than that among patients without recurrence (median, 47 vs 22 mm/hr, P < 0.001), supporting the accuracy of recurrence status. 

Table R4. The ESR value after treatment completion for patients with PVO, by status of 6-month recurrence.

Variables, median (IQR) No Recurrence

(N = 323) Recurrence

(N = 53) p-value

Available N 202 (62.5%) 48 (90.6%) 

ESR (mm/hr) 22.0 (11.0-47.3) 47.0 (25.0-86.0) < 0.001

Days from the end of antibiotic 

treatment to the closest ESR 13.0 (3.0-26.8) 13.0 (6.0-31.8) 0.397

Revised content:

Page 7, Line 135: In the MATERIALS and METHODS section under Covariables and outcomes, we added the following sentences to clarify the definition of recurrence: “The EMRs were reviewed by two Infectious Diseases Specialists (CYC and WSC) to determine patients’ delayed operation and recurrence status. …. Recurrence was defined as any recurrent symptoms and signs (e.g., fever or pain on the affected site accompanied with abnormal image study or increased inflammatory markers, in the absence of other causes) within 6 months after the completion of the initial antibiotic treatment and received another course of antibiotic treatment based on clinician’s decision.“

5. “if a surgical intervention was performed> 2 weeks after PVO diagnosis, it was defined as a delayed operation” Please, could you explain the meaning of this sentence in more detail?. I guess do you refer to a surgical intervention to treat the infection, although it could also refer to corrective surgery (by sequelae)…. If it is a surgical treatment of the infection, does it refer to patients who had an initial indication of surgery, but was it performed “late”? Or who were operated late for lack of initial clinical and / or radiological improvement with antibiotic treatment (but without apparent initial surgical indication? The interpretation of the surgical variable can be very different (“consequence” of bad evolution or “cause” of bad evolution due to the non-early performance)

Response: 

In the present study, two Infectious Disease Specialists (CCY and WSC) had reviewed the medical records for all patients and defined patients receiving delayed surgery if they had initial indications for surgical intervention (e.g., neural compression, spinal instability, or presence of epidural or paravertebral abscesses) at the time of PVO diagnosis but were operated (e.g., discectomy, laminectomy) later than 2 weeks after the diagnosis of PVO. Therefore, the delayed surgery was not the consequence of worse ESR response or insufficient treatment.

Revised content:

Page 7, Line 135: In the MATERIALS and METHODS section under Covariables and outcomes, we edited the following sentences: “The EMRs were reviewed by two Infectious Diseases Specialists (CYC and WSC) to determine patients’ delayed operation and recurrence status. We defined surgical procedures performed within 2 weeks after PVO diagnosis as immediate operations. Patients receiving delayed surgery were defined if they had initial indications for surgical intervention (e.g., neural compression, spinal instability, or presence of epidural or paravertebral abscesses) at the time of PVO diagnosis but were operated (e.g., discectomy, laminectomy) later than 2 weeks after the diagnosis of PVO.”

6. The statistical analysis does not specifically mention the performance of trend analysis, which is after that cited in several sections.

Response: We thank Reviewer for pointing out this issue. The test for linear trend from ESR Group 1, 2, to 3 was evaluated by modeling the categorical ESR trajectories as a continuous variable in simple linear regression models. We have added a few sentences in the METHOD section to clarify the trend analysis. 

Revised content:

Page 9, Line 183: In MATERIALS and METHODS section under Statistical analysis, we added the following sentence: “The test for linear trend from ESR Group 1, 2, to 3 was evaluated by modeling the categorical ESR trajectories as a continuous variable in simple linear regression models.”

Results:

7. The reference to Figure 1 should be at the end of line 176

Response: We have edited the manuscript according to the Reviewer 2’s suggestion.

Revised content:

Page 11, Line 205: In RESULTS section, we moved the Figure 1 reference to the end of the first sentence: “Of the 502 patients with a discharge diagnosis of vertebral osteomyelitis, 401 patients met the current inclusion criteria and were included in the final analysis (Figure 1).”

8. Figure 2: it is not necessary to repeat the number (1, 2 or 3) on each line

Response: Thank you. We have modified the Figure 2 accordingly.

Revised content: In Figure 2, we’ve replaced the numbers (1, 2, 3) with circles, triangles, and squares. 

9. Line 193: “P for trend” à the meaning is not explained neither has been previously mentioned the use of this statistics; so, it should be clarified

Response: Thank you. In Table 1, we have added an additional column for “p value for trend”, which was derived from the test for linear trend by modeling the categorical ESR trajectories as a continuous variable in simple linear regression models.

Revised content: 

Page 9, Line 183: In MATERIALS and METHODS under Statistical analysis, we have added detailed information of trend analysis: “The test for linear trend was evaluated by modeling the categorical ESR trajectories as a continuous variable in simple linear regression models.” 

10. Table 1: the meaning of “NA” is not defined. In some cases there is a result of "p for trend" and in others apparently not (that is, in these cases the "p" would only reflect that there is some difference between the 3 groups?); this issue should be clarified

Response: Thank you. In Table 1, we have added an additional column for “p value for trend” and remove the “NA”.

Revised content: Revised Table 1. 

11. It would be interesting to know if group 3 had overall more comorbidities (or any comorbidity) compared to the other groups, apart from the specific comorbidities mentioned (lines 192-194)

Response: Thank you. We have provided additional comorbidities based on Charlson’s comorbidity [7] and also the Charlson’s comorbidity index (CCI) in Table 1 and have added one sentence in the RESULTS section. The original comorbidities we presented (diabetes, IV drug use, ESRD, and malignancy) were significantly associated with ESR trajectory groups and no additional comorbidity was significant (Table 1). The mean CCI was 1.09 ± 1.69 for ESR group 1, 1.47 ± 1.91 for ESR group 2, and 2.04 ± 2.43 for ESR group 3 (p for trend < 0.001). Because CCI was significantly different across the three ESR groups, we added CCI >= 3 as a new predicting variable in the multivariable analysis for ESR groups (Table 3), prolonged treatment duration (Table 4), and recurrence (Table 5). We have updated all the data in Table 4, 5, 6 and in Supplementary Table 4. The updated ORs were similar to the ORs from the original models. Please see Figure R1 for the comparison. In addition, we re-draw Figure 4, Supplementary Figure 3, 4, and 5 because these figures involve the use of the full models.

 

Figure R1. Results from the original and the updated multivariable models.

(A) Original Table 3 vs. Updated Table 3

(B) Original Table 4 vs. Updated Table 4

(C) Original Table 5 vs. Updated Table 5

Revised content:

1) Page 11, Line 226: In the RESULTS section under 2nd paragraph, we added one sentence to describe Charlson’s comorbidity score: “The mean Charlson’s comorbidity index was 1.09 ± 1.69 for Group 1, 1.47 ± 1.91 for Group 2, and 2.04 ± 2.43 for Group 3, which was significantly different (P for trend < 0.001).”

2) Page 14, Line 287 & Line 293: In the RESULTS section under the last paragraph, we updated the c-statistic for Model 2-3: “The C statistic of the full model (Model 2-3) indicated moderate-to-good discrimination (0.75; 95% CI, 0.70 to 0.81)”. “…their discriminative performance was improved with the addition of clinical information (Model 3-3; C = 0.69; 95% CI, 0.61 to 0.78)…”

3) Table 1: We added more comorbidities and the Charlson’s comorbidity index.

4) Table 3, 4, 5 & Supplementary Table 4: We added the Charlson’s comorbidity in the Model 1-2 and 1-3 (Table 3), Model 2-3 (Table 4), Model 3-3 (Table 5) and Model S1-2 and S1-3 (Supplementary Table 4) and we updated all the data.

5) Figure 4, Supplementary Figure 3, 4, and 5: We re-draw the ROC curve for prolonged treatment duration (Figure 4) and for recurrence (Supplementary Figure 3). Also, we re-did the two calibration plots for prolonged treatment duration (Supplementary Figure 4) and for recurrence (Supplementary Figure 5). 

12. Were there patients with a simultaneous diagnosis of endocarditis? Was this possibility assessed regularly among patients with infection or bacteraemia documented by gram-positive cocci?

Response: Thank you for this important question. Searching for evidence of infective endocarditis is not a routine at our institution. After carefully screening whether our study population met modified Duke criteria within the 30-day window prior to the diagnosis of PVO, we identified only 3 patients (0.75%) had infective endocarditis (IE). All of them had been treated adequately. In addition, only 1 of these 3 patients were classified in the ESR Group 3 that had high initial ESR and responded slow. Thus, we believe that work-up bias and information bias are unlikely and excluding 3 patients with recent IE would not change our findings.

Revised content: None.

13. Could it be analyzed if the indication for immediate surgical treatment changed over time? A percentage of 50% seems very high for current practice

Response: We thank the reviewer for this critical concern. In our study, as high as 51% of PVO patients received immediate surgical treatment, possibly due to the high proportion (80%) of patient with abscess that required immediate surgery for local debridement or decompression (Table 1). The indications for surgical intervention for PVO are well-known and they are unlikely to be changed in the past 13 years at our hospital. The proportion of patients receiving immediate surgery or the proportion with abscess did not change much during the study period (Table R5). However, in order to keep a good rapport with patients who have indications of surgical intervention, the decision of operation does not depend solely on in-charge surgeon but also on patients and their family. For regulations of health insurance system and regional cultures of Taiwan, most patients and their family will follow the suggestions of in-charge surgeon.

Table R5. Number of PVO patients who received immediate surgical treatment, by PVO diagnosis year.

Year No. of PVO patients No. of PVO patients with immediate surgery (%) No. of PVO patients with abscess (%)

2002 1 1 (100) 1 (100)

2003 13 5 (38.5) 8 (61.5)

2004 25 15 (60) 20 (80.0)

2005 21 9 (42.9) 17 (81.0)

2006 35 21 (60) 29 (82.9)

2007 31 18 (58.1) 22 (71.0)

2008 36 12 (33.3) 31 (86.1)

2009 40 20 (50) 27 (67.5)

2010 35 17 (48.6) 26 (74.3)

2011 58 35 (60.3) 49 (84.5)

2012 49 20 (40.8) 43 (87.8)

2013 36 24 (66.7) 32 (88.9)

2014 21 9 (42.9) 16 (76.2)

Revised content: None.

14. The 3 defined trajectory pattern show evident differences in the starting point and evolution of the ESR. However, finally, the prognostic differences seem to be found only between two groups (3 versus 1 + 2 or 1 with respect to 2 + 3 groups). The text and tables 3-5 do not quite clarify this point. In this situation, perhaps after a first descriptive analysis of the 3 groups, the most sophisticated (multivariate) analysis of the outcomes could be done considering the 2 groups that present prognostic differences in the analyzed outcomes; this, in addition, would increase the statistical power to be able to detect differences.

Response: We thank Reviewer for the suggestion. In Table 1 in the main text, the proportion of prolonged treatment was 32%, 59%, and 55.5% for ESR Group 1, 2, 3, respectively. The recurrence was 7.7%, 15.0%, and 19.0% for ESR Group 1, 2, 3, respectively. Since the proportion of prolonged treatment and the proportion of recurrence was comparable for Group 2 and 3, it is reasonable to combine Group 2 and 3 in the multivariable prediction model of ESR group (Table R6). In the full model, initial ESR and ESR-AD remained to be the only two significant predictors for ESR groups, same as the results from the modeling of 3 ESR groups (Table 3).

Table R6. Prediction of erythrocyte sedimentation rate (ESR) trajectory group (Groups 1 vs. 2+3) through logistic regression based on ESR within 4 weeks of pyogenic vertebral osteomyelitis diagnosis.

Variables Crude OR

(95% CI) Model 1-1 (AIC: 252.1) b Model 1-2 (AIC: 238.9) c Model 1-3 (AIC: 240.9) d

　 β Proportional OR

(95% CI) β Proportional OR

(95% CI) β Proportional OR

(95% CI)

Initial ESR, mm/h 1.05 (1.04, 1.06) 0.089 1.09 (1.07, 1.12) 0.090 1.09 (1.07, 1.12) 0.090 1.09 (1.07, 1.12)

ESR – AD ≥ −9 mm/h a 2.46 (1.56, 3.93) 3.633 37.84 (15.37, 105.48) 3.682 39.72 (14.84, 123.1) 3.679 39.59 (14.76, 122.9)

Age, year 1.02 (1.00, 1.04) 0.002 1.00 (0.97, 1.03) 0.003 1.00 (0.97, 1.03)

Male 0.65 (0.41, 1.03) -0.680 0.51 (0.23, 1.10) -0.683 0.51 (0.23, 1.1)

Diabetes mellitus 3.13 (1.91, 5.27) 0.686 1.99 (0.92, 4.38) 0.683 1.98 (0.92, 4.38)

ESRD 4.78 (1.66, 20.2) 0.424 1.53 (0.24, 12.85) 0.413 1.51 (0.24, 12.85)

Malignancy 2.82 (1.16, 8.43) 1.323 3.75 (0.66, 23.21) 1.318 3.74 (0.66, 23.19)

eGFR, mL/min/1.73 m² 0.98 (0.98, 0.99) -0.010 0.99 (0.97, 1.01) -0.010 0.99 (0.97, 1.01)

CCI ≥ 3 2.77 (1.56, 5.21) -0.224 0.80 (0.28, 2.28) -0.226 0.80 (0.28, 2.27)

Mixed bacteria 2.90 (1.10, 10.0) 0.061 1.08 (0.23, 5.59)

C statistic (95% CI) 0.91 (0.87 - 0.94) 0.92 (0.89 - 0.95) 0.92 (0.89 - 0.95)

Abbreviations: AD, absolute difference; AIC, Akaike Information Criterion; CCI, Charlson’s comorbidities index; CI, confidence interval; eGFR, estimated glomerular filtration rate; ESR, erythrocyte sedimentation rate; ESRD, end-stage renal disease; OR, odds ratio; PVO, pyogenic vertebral osteomyelitis.

a. We used the median value of ESR – AD as the cutoff value.

b. In Model 1-1, initial ESR and ESR – AD were included as predicting variables and the ESR groups (1 vs. 2+3) as the dichotomous response variable.

c. In Model 1-2, initial ESR, ESR – AD, demographic information (age, gender), baseline comorbidities (diabetes, ESRD, malignancy, and CCI) and baseline eGFR were included as predicting variables and he ESR groups (1 vs. 2+3) as the dichotomous response variable.

d. In Model 1-3 (full model), initial ESR, ESR – AD, demographic information (age, gender), baseline comorbidities (diabetes, ESRD, malignancy, and CCI), baseline eGFR, and polymicrobial infection were included as predicting variables and the ESR groups (1 vs. 2+3) as the dichotomous response variable.

Revised content:

Supplementary Table 4: We added the multivariable prediction model for ESR group 1 vs ESR group 2+3 in the Supplementary Table 4.

15. In tables 3-5, the different models performed should be explained at the bottom of the table (the tables must be self-explanatory).

Response: Thank you. We have added relevant information to facilitate the readability of these tables. In the top first cell of the first column, we added the column title “Variables” to help readers identify the sequential changes in the adjusted co-variables of all statistical models. In addition, we added footnotes under Table 3, 4, and 5.

Revised content: In Table 3, 4, and 5, we added footnotes to describe the predicting variables and the response variable for each model.

16. In the multivariate models described in Tables 3-5, how have been decided which variables to include? It seems somewhat arbitrary ... Have all the variables that could influence the prognosis been considered (particularly those kwown variables increasing the risk of recurrence?

Response: Thank you for this important question. These variables were adjusted because they are statistically significant in the univariable analysis (mixed culture, DM, ESRD, malignancy; Table 1) or clinically relevant (CRP, WBC, eGFR). Since poor prognosis is a multifactorial event, we should identify all potential confounders that may be involved in the causal pathway. The best way to illustrate a causal pathway is using a directed acyclic graph or DAG. We provided a DAG for readers to understand the potential confounding induced by the variables we adjusted for (Supplementary Figure 1B; Figure R2). 

The main exposure of interest is “4-week ESR variability” and the main outcome of interest is “Poor prognosis” (i.e., prolonged treatment or recurrence). By blocking the backdoor path “4-week ESR variability – other inflammatory markers – comorbidities – age, gender”, the independent effect of 4 week-ESR variability on prognosis can be estimated (Figure R2). We addressed this issue in the multivariable analyses (Table 4, 5).

Figure R2. Directed acyclic graph (DAG) for the causal relationship between covariates and poor prognosis among patients with PVO - adjusted for age, gender, comorbidities (i.e., diabetes, ESRD, malignancy, Charlson’s Comorbidity Index), other inflammatory marker (i.e., CRP), the presence of abscess, and delayed operation.

Revised content:

1) Page 9, Line 191: In MATERIALS and METHODS under Statistical analysis, we updated the following sentences: “Furthermore, variables that were statistically significant in the univariable analysis (polymicrobial culture, DM, ESRD, malignancy, CCI; Table 1) or were clinically relevant (CRP, WBC, eGFR) were included in the multivariable analysis. By incorporating the first-4-week ESR variabilities, initial CRP, and other potential confounders, we built several models using logistic regression to predict prolonged PVO treatment (≥ 12 weeks) and recurrence. In addition, we proposed a directed acyclic graph (DAG) to illustrate the potential causal relationship between the first-4-week ESR variabilities and poor prognosis (Supplementary Figure 1).”

2) Supplementary Figure 1: We added a DAG to illustrate the causal relationship between covariates and poor prognosis.

17. I suppose that it has been decided to analyze the ESR because it presents a better predictive profile of the evolution of vertebral osteomyelitis. However, I would like to know what the situation in the case of the CRP and why the analysis of this marker was ruled out. In principle, it is thought that CRP could be the parameter of choice. The evolution of leukocytes, CRP and ESR in a single graph could be of interest; it could also be considered to analyze the mentioned markers according to the current 3 defined trajectory patterns. In supplementary figure 1, it is clear that the division into 3 groups of patients in relation to CRP is not useful; in this sense, surgical treatment should probably have an important role that makes interpretation difficult in these patients (in which CRP should have a significant increase in relation to surgery that distorts the overall results); maybe the CRP analysis could be considered excluding surgery patients or excluding CRP increases related to surgery.

Response: 

1) In the initial analysis of GBTM, we did analyze serial measurements of ESR, C-reactive protein (CRP), and white blood cell counts (WBCs) among patients with PVO (Figure 2 and Supplementary Figure 2). Only ESR trajectories distinctly demonstrated different clinical courses among patients with PVO. The CRP level for 79.4% of our study population returned to less than 5 mg/dL in the second week after PVO diagnosis (Group 2, Supplementary Figure 2A). In addition, WBC trajectories did not show significantly different pattern and 73.5% of our study population had flat pattern of WBC (Group 1, Supplementary Figure 2B). Therefore, we focused on ESR as the main marker to predict the disease course of PVO. 

2) To address Reviewer 2’s comment, we have plotted the patterns of CRP and WBC throughout the first 6 months of PVO course for each of the current 3 defined ESR groups (Figure R3A). Patients with initial-high ESR (ESR Group 2 and 3) tend to have initial-high CRP as well. However, the CRP declined consistently across three ESR groups with similar magnitude during the first 1-2 weeks and after that. The correlation between ESR and CRP then disappeared after the fourth month of treatment. As for WBC pattern (Figure R3B), there’s no distinct difference across the 3 ESR groups since the second month of treatment. Therefore, in treating patients with PVO, ESR would be a superior follow-up marker with a noticeable resolution in guiding treatment courses.

Figure R3. The pattern of C-reactive protein (CRP) and the pattern of white blood cell count (WBC) within the first 6 months of PVO diagnosis, stratified by the three ESR trajectory groups.

(A) CRP pattern by ESR group

(B) WBC pattern by ESR group

3) We agree with Reviewer 2 that surgery per se may interfere the level of CRP. For instance, in an early study done by Larsson et al., they examined the CRP levels after elective orthopedic surgery and found that the CRP increased and peaked on the second or the third day after the surgery, followed by dropping to normal within 21 days postoperatively [8]. However, this finding may not extrapolate to surgical intervention for infectious diseases. In our study, the initial level of CRP was the highest for patients receiving operation (either immediate or delayed) and the CRP remained higher for patients with delayed operation within the first 4 weeks (Figure R4). However, the CRP level decreased to similar level for all patients within 5 weeks after the PVO diagnosis, regardless of their operation status. Therefore, this phenomenon indirectly supports the use of ESR in predicting the clinical course of PVO. We think that excluding patients undergone surgery while analyzing the CRP patterns would not be appropriate because 271 patients (67.6%) in our study population would be excluded.

Figure R4. The pattern of C-reactive protein (CRP) within the first 6 months of PVO diagnosis, stratified by the surgical treatment.

Revised content:

1) Page 15, Line 304: In the DISCUSSION section under the first paragraph, we added: “Additionally, WBC trajectories did not show significantly different pattern and 73.5% of our study population had flat pattern of WBC (Group 1, Supplementary Figure 2B). “

2) Page 15, Line 316: In the DISCUSSION section under the first paragraph, we added: “In fact, the CRP level for 79.4% of our study population returned to less than 5 mg/dL in the second week after PVO diagnosis (Group 2, Supplementary Figure 2A).”

18. Figure 3 is not very intuitive

Response: We agree with the reviewer that this probability graph is not intuitive for most clinicians; however, we humbly ask to keep this graph for readers who are interested in applying the findings of medical research in real practice. First, we have provided case studies in the plot to illustrate how to interpret this probability graph. Second, in the era of electronic medical records (EMRs), it is easy to embed this probability graph into EMR system. We have specifically highlighted (black-dashed line) the two hypothetical patients in the revised Figure 3 to improve its readability.

Revised content:

Figure 3: We have specifically highlighted (black-dashed line) the two hypothetical patients in the figure to improve its readability.

Discussion

19. Lines 275: “The inflammatory marker CRP returns to its normal level within 1 week”. Maybe it means "may" (or something similar) returns?

Response: Thank you. We have toned down the expression and modified this sentence accordingly. 

Revised content:

Page 15, Line 312: We have edited the following sentence: “The inflammatory marker CRP may return to its normal level within 1 week.”

 

REFERENCES

1. Lin Z, Vasudevan A, Tambyah PA. Use of erythrocyte sedimentation rate and C-reactive protein to predict osteomyelitis recurrence. J Orthop Surg (Hong Kong). 2016;24(1):77-83. doi: 10.1177/230949901602400118. PubMed PMID: 27122518.

2. Chang WS, Ho MW, Lin PC, Ho CM, Chou CH, Lu MC, et al. Clinical characteristics, treatments, and outcomes of hematogenous pyogenic vertebral osteomyelitis, 12-year experience from a tertiary hospital in central Taiwan. J Microbiol Immunol Infect. 2018;51(2):235-42. doi: 10.1016/j.jmii.2017.08.002. PubMed PMID: 28847713.

3. Nickerson EK, Sinha R. Vertebral osteomyelitis in adults: an update. Br Med Bull. 2016;117(1):121-38. doi: 10.1093/bmb/ldw003. PubMed PMID: 26872859.

4. Segreto FA, Beyer GA, Grieco P, Horn SR, Bortz CA, Jalai CM, Passias PG, Paulino CB, Diebo BG. Vertebral Osteomyelitis: A Comparison of Associated Outcomes in Early Versus Delayed Surgical Treatment. Int J Spine Surg. 2018 Dec 21;12(6):703-712. doi: 10.14444/5088. eCollection 2018 Dec. 

5. Adogwa O, Karikari IO, Carr KR, Krucoff M, Ajay D, Fatemi P, Perez EL, Cheng JS, Bagley CA, Isaacs RE. Spontaneous spinal epidural abscess in patients 50 years of age and older: a 15-year institutional perspective and review of the literature: clinical article. J Neurosurg Spine. 2014 Mar;20(3):344-9. doi: 10.3171/2013.11.SPINE13527. Epub 2013 Dec 20.

6. Karikari IO, Powers CJ, Reynolds RM, Mehta AI, Isaacs RE. Management of a spontaneous spinal epidural abscess: a single-center 10-year experience. Neurosurgery. 2009 Nov;65(5):919-23; discussion 923-4. doi: 10.1227/01.NEU.0000356972.97356.C5.

7. Quan H, Sundararajan V, Halfon P, Fong A, Burnand B, Luthi JC, Saunders LD, Beck CA, Feasby TE, Ghali WA. Coding algorithms for defining comorbidities in ICD-9-CM and ICD-10 administrative data. Med Care. 2005 Nov;43(11):1130-9.

8. Larsson S, Thelander U, Friberg S. C-reactive protein (CRP) levels after elective orthopedic surgery. Clin Orthop Relat Res. 1992 Feb;(275):237-42.

---

## [Editor Report · Decision Letter 1]

18 Nov 2019

First-4-Week Erythrocyte Sedimentation Rate Variability Predicts Erythrocyte Sedimentation Rate Trajectories and Clinical Course among Patients with Pyogenic Vertebral Osteomyelitis

PONE-D-19-15739R1

Dear Dr. Kuo,

We are pleased to inform you that your manuscript has been judged scientifically suitable for publication and will be formally accepted for publication once it complies with all outstanding technical requirements.

With kind regards,

Daniel Pérez-Prieto, PhD

Academic Editor

PLOS ONE
---

## [Editor Report · Acceptance letter]

21 Nov 2019

PONE-D-19-15739R1 

First-4-Week Erythrocyte Sedimentation Rate Variability Predicts Erythrocyte Sedimentation Rate Trajectories and Clinical Course among Patients with Pyogenic Vertebral Osteomyelitis 

Dear Dr. Kuo:

I am pleased to inform you that your manuscript has been deemed suitable for publication in PLOS ONE. Congratulations! Your manuscript is now with our production department. 

With kind regards,

on behalf of

Dr. Daniel Pérez-Prieto 

Academic Editor

PLOS ONE